# Tandem gene duplications contributed to high-level azole resistance in a rapidly expanding *Candida tropicalis* population

Xin Fan[1,2,10], Rong-Chen Dai [2,3,10], Shu Zhang[3,4], Yuan-Yuan Geng[3,4], Mei Kang[5], Da-Wen Guo[6], Ya-Ning Mei[7], Yu-Hong Pan[8], Zi-Yong Sun[9], Ying-Chun Xu[2] ✉, Jie Gong [3,4] ✉ & Meng Xiao [2] ✉

Invasive diseases caused by the globally distributed commensal yeast *Candida tropicalis* are associated with mortality rates of greater than 50%. Notable increases of azole resistance have been observed in this species, particularly within Asia-Pacific regions. Here, we carried out a genetic population study on 1571 global *C. tropicalis* isolates using multilocus sequence typing (MLST). In addition, whole-genome sequencing (WGS) analysis was conducted on 629 of these strains, comprising 448 clinical invasive strains obtained in this study and 181 genomes sourced from public databases. We found that MLST clade 4 is the predominant azole-resistant clone. WGS analyses demonstrated that dramatically increasing rates of azole resistance are associated with a rapid expansion of cluster AZR, a sublineage of clade 4. Cluster AZR isolates exhibited a distinct high-level azole resistance, which was induced by tandem duplications of the *ERG11*[A395T] gene allele. *Ty3/gypsy*-like retrotransposons were found to be highly enriched in this population. The alarming expansion of *C. tropicalis* cluster AZR population underscores the urgent need for strategies against growing threats of antifungal resistance.

The fungal kingdom has been estimated to consist of up to 6 million species that are distributed across the globe[1,2]. As a key human-associated fungal genus, *Candida* includes species that are important components of the human commensal microbiome, but many species are also opportunistic pathogens that cause a broad range of human diseases, from cutaneous infections to lethal invasive candidiases[3,4].

Cases of invasive candidiasis are of particular note, as this disease is estimated to be associated with mortality rates that exceed 50%[2,3,5]. Although *Candida albicans* remains the most common infection-causing *Candida* species, other pathogenic *Candida* species have increasingly been isolated over the past several decades[3,6]. Among these species, *C. tropicalis* is the second- to fourth-most common

[1]Department of Infectious Diseases and Clinical Microbiology, Beijing Institute of Respiratory Medicine and Beijing Chao-Yang Hospital, Capital Medical University, Beijing 100020, China. [2]Beijing Key Laboratory for Mechanisms Research and Precision Diagnosis of Invasive Fungal Diseases, Department of Laboratory Medicine, State Key Laboratory of Complex Severe and Rare Diseases, Peking Union Medical College Hospital, Chinese Academy of Medical Sciences, Beijing 100730, China. [3]National Key Laboratory of Intelligent Tracking and Forecasting for Infectious Diseases, National Institute for Communicable Disease Control and Prevention, Chinese Center for Disease Control and Prevention, Beijing 102206, China. [4]Peking University First Hospital - National Institute for Communicable Disease Control and Prevention Joint Laboratory of Pathogenic Fungi, Beijing 102206, China. [5]Department of Laboratory Medicine, West China Hospital, Sichuan University, Chengdu 610041 Sichuan, China. [6]Department of Clinical Laboratory, First Affiliated Hospital of Harbin Medical University, Harbin 150001 Heilongjiang, China. [7]Department of Clinical Laboratory, Jiangsu Province Hospital, Nanjing 210029 Jiangsu, China. [8]Department of Clinical Laboratory, Fujian Medical University Union Hospital, Fuzhou 350001 Fujian, China. [9]Department of Clinical Laboratory, Tongji Hospital, Tongji Medical College of Huazhong University of Science and Technology, Wuhan 430030 Hubei, China. [10]These authors contributed equally: Xin Fan, Rong-Chen Dai. ✉e-mail: xycpumch@139.com; gongjie@icdc.cn; cjtcxiaomeng@aliyun.com

cause of candidiasis worldwide, and infections caused by *C. tropicalis* tend to be associated with worse outcomes than those associated with *C. albicans*[3,5]. Moreover, while other commonly seen *Candida* species, such as *C. albicans* and *C. parapsilosis*, remain >90% susceptible to all antifungal agents, azole resistance in *C. tropicalis* has become a worrisome problem. This issue is especially striking in the Asia-Pacific region. *C. tropicalis* is more prevalent in the Asia-Pacific and Latin America regions[6,7], and rates of azole resistance among cases of invasive candidiasis have reached as much as 16–30% in countries like Australia and China[6–8].

Notably, associations between specific genetic lineages and decreased antifungal susceptibility have been revealed for several *Candida* species, and the spread of drug-resistant clones has posed difficult clinical challenges[7,9,10]. By using various typing schemes, such as multilocus sequence typing (MLST) and microsatellite assays, previous studies have demonstrated that *C. tropicalis* is associated with high genetic diversity, but most azole-resistant strains, especially those identified in the Asia-Pacific region, cluster in a few genetically-related groups[11–13]. Moreover, a longitudinal study in Taiwan using the MLST scheme has identified a shift of major resistant clones over time[13].

In recent years, the application of whole-genome sequencing (WGS) has allowed for a more detailed understanding of population structure, evolution, and geographic dissemination of different microbial pathogens. To date, there have been two large population genomics studies carried out on *C. tropicalis*. In one such study, O'Brien et al. studied 77 *C. tropicalis* isolates from clinical and environmental sources collected worldwide, and they determined that hybridization has been common in the evolutionary history of *C. tropicalis*[12]. The second study, by Keighley et al., used more than 80 clinical isolates from Australia and Singapore and identified a certain clonal complex (CC, i.e., MLST CC2) and particular gene polymorphisms that were associated with reduced azole susceptibility[7]. However, these two previous studies incorporated only a limited number of azole-resistant strains, and the potential connections between WGS phylogenetic clusters and geographic variations as well as dynamic changes of antifungal resistance remain unclear.

Further investigation of such connections may lead to a deeper understanding of the mechanisms responsible for the emergence and spread of antifungal resistance. In *C. tropicalis*, it has been well established that certain *ERG11* gene mutations that lead to key amino acid substitutions in the azole target Erg11p (e.g., *ERG11* gene mutation A395T conferring a key substitution Y132F) represent major mechanisms responsible for azole resistance[14,15]. Other mechanisms associated with azole resistance in *C. tropicalis* include the induction of *ERG11* overexpression by overexpression or missense mutations of the transcription factor-encoding gene *UPC2*[7,16]. Similarly, resistance can be induced by the upregulation of expression of drug efflux transporters, such as those of the Cdr and Mdr families, due to mutations in relevant transcription factor-encoding genes (e.g., *TAC1* and *MRR1*)[14,15]. These resistance mechanisms are found almost universally in the *Candida* genus[3,14], but an open question is whether unique mechanisms have led to the recent dramatic increases in azole-resistance rates, which have been distinctively observed in *C. tropicalis* but not in other *Candida* species.

Copy number variations, or CNVs, involve duplications or deletions of chromosomal segments and are major contributors to genome variabilities in diverse organisms, from microbes to plants and animals[17–19]. Some CNVs appear recurrently as either tandem duplications, in which one or two genes duplicate in tandem; segment duplications, in which a few genes to entire chromosomes duplicate; or polyploidies, which is synonymous with whole-genome duplication[17,19]. Other CNVs show non-recurrent end-points in structure; these are called non-recurrent CNVs[17]. Importantly, CNVs may change the levels of gene expression and can result in phenotypic effects[17]. In humans, CNVs are associated with a variety of diseases, including cancer, mental illnesses and autoimmune diseases[17,18].

On the other hand, alterations in gene copy numbers may also bring organisms adaptive advantages, such as enhancing antimicrobial resistance or tolerance in microorganisms[14,20,21]. In *C. albicans* and *C. tropicalis*, there have been reports that *ERG11* gene CNVs have led to decreased azole susceptibility of strains; all of the previously identified cases resulted from aneuploidies of large chromosome segments that contained *ERG11* gene coding domain sequences[22,23].

In the present study, we carried out a large genetic population analysis of 1571 *C. tropicalis* global strains, comprising 629 isolates with complete genomes and an additional 942 isolates with MLST information. It was found that increases in rates of azole resistance are correlated with the expansion of a specific WGS phylogenetic sublineage, which was named cluster AZR. Of note, in addition to carrying A395T mutations, tandem gene duplication CNVs of the *ERG11* gene were identified for the first time in *C. tropicalis*; these CNVs represent a unique feature of cluster AZR isolates and were associated with high-level azole resistance (fluconazole ≥ 256 mg/L). In addition, class-I retrotransposons were found to be more enriched in cluster AZR isolates, which indicated a higher potential of genome plasticity within this population. Altogether, our findings provide new insights to explain the rapid expansion of azole resistance in *C. tropicalis*, and they allowed us to identify a key target population that requires more effective interventions.

## Results

### Distinct geographic distribution of *C. tropicalis* MLST clades are associated with intercontinental variations of azole resistance

We analyzed MLST characteristics of 1571 *C. tropicalis* isolates distributed worldwide (Figs. 1 and 2). The collection was composed of 629 *C. tropicalis* isolates with WGS results, including 181 international strains that were described in nine previous studies, and 448 strains causing human invasive fungal diseases (IFDs) in China that were sequenced in this study (Supplementary Data 1 and 2). The collection also included an additional 942 isolates whose information was obtained from published literature and the PubMLST database (http://www.pubmlst.net) (Supplementary Data 1).

A total of 726 diploid sequence types (DSTs) were identified among 1571 isolates, including 201 novel DSTs that were not recorded in the PubMLST database (Supplementary Data 1). Overall, 73.8% (1159/1571) of isolates could be assigned to 29 MLST clades, including 21 that were proposed previously[13] and eight new designated in this study (Fig. 2, Supplementary Data 2). MLST clade 4 was the most common clade identified (15.8% of 1571 isolates), followed by clade 5 (10.1%) (Figs. 2 and 3a). These two clades were also predominant in Asia, where clade 4 and clade 5 included 19.3% and 11.7% of 1270 isolates collected, respectively (Fig. 3a). Geographic variations were observed across Europe, Oceania, and North America, where clade N6 (16.3% [26/160]), clade 3 (17.6% [13/74]) and clade 2 (15.8% [9/57]) were the most prevalent, respectively (Fig. 3a).

The overall rates of fluconazole resistance were 31.0% (*n* = 487) in the 1571 isolates. Of note, both MLST clade 4 and clade 5 exhibited significantly higher rates of azole resistance (93.1% and 31.0%, respectively) than did other clades (17.8% overall, both *p* < 0.001) (Figs. 2 and 3b). Clades 4 and 5 were the predominant azole-resistant genotypes globally, as they included 49.6% and 8.9% of all resistant isolates, respectively. The intercontinental variations of *C. tropicalis* clades, along with divergent antifungal resistant profiles among different clades, were considered to be associated with the significantly higher azole resistance rate found in Asia (35.4%) compared to other continents (13.0% overall) (Fig. 3c).

### A recent shift of the predominant azole-resistant *C. tropicalis* lineage to MLST clade 4

The earliest strains in the *C. tropicalis* MLST database can be traced back to the 1970s, and the first fluconazole-resistant strain recorded

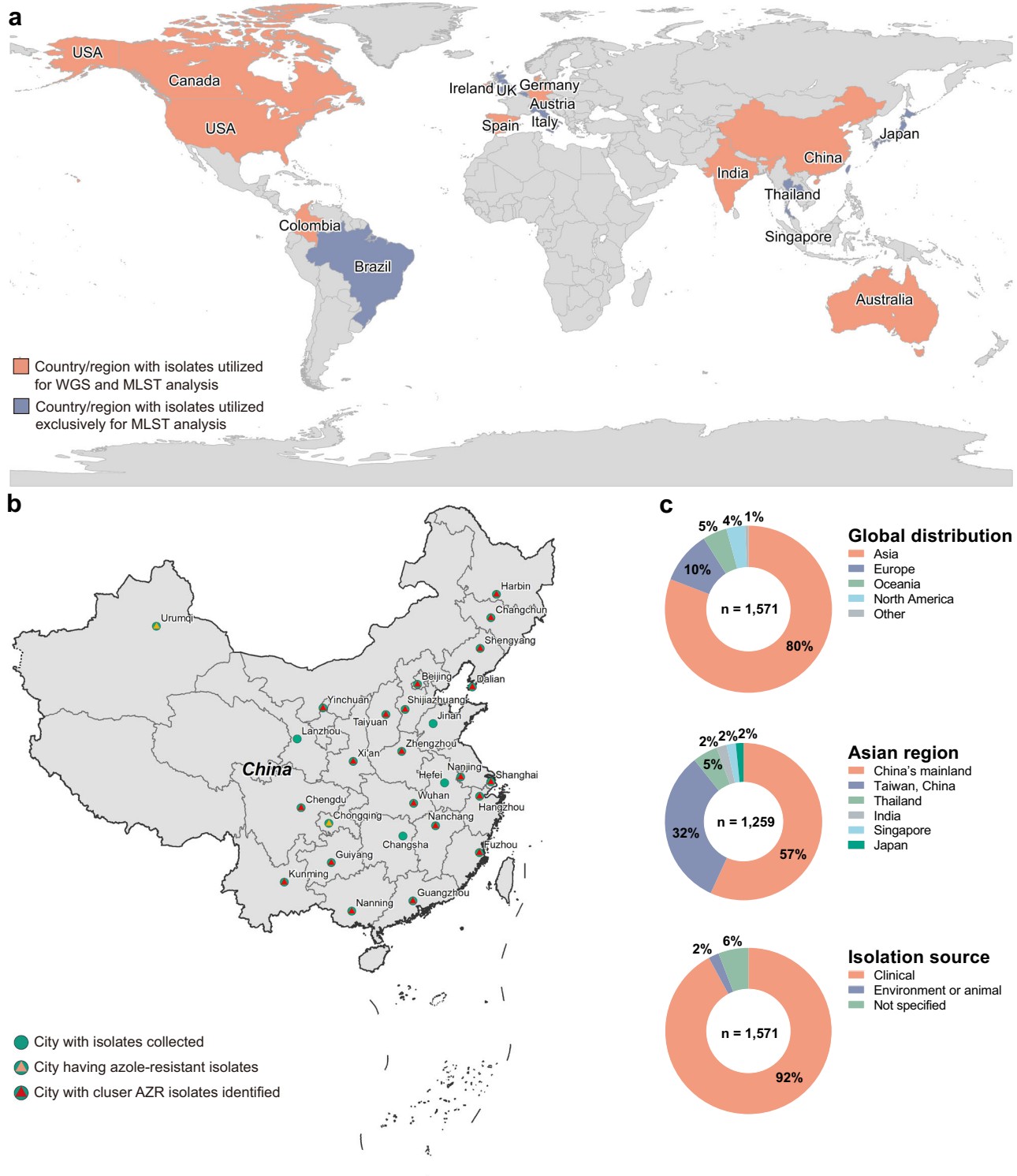

**Fig. 1 | Geographic distribution of *C. tropicalis* isolates involved in this study. a** Global geographic origins of 1571 *C. tropicalis* isolates used for multilocus sequence typing (MLST) and whole-genome sequencing (WGS) analyses in this study by country. The collection of isolates included (A) 629 *C. tropicalis* isolates with WGS results, including 181 international strains that were described in nine previous studies, and 448 strains causing human invasive fungal diseases (IFDs) in China (country labeled in orange, Supplementary Data 1), and (B) 942 isolates from previous publications or an online MLST database with MLST information (country labeled in blue, Supplementary Data 2). **b** Geographic distribution of 448 *C. tropicalis* isolates collected in China by city (labeled with green circles). Cities from which azole-resistant strains and phylogenetic cluster AZR isolates were collected are labeled with orange and red triangles embedded in the green circles, respectively. **c** Detailed information on the proportion of *C. tropicalis* isolates collected, including global distribution for 1571 isolates by continents (1c top), distribution for 1259 isolates collected in Asia by region (1c middle), and strain isolation source (1c bottom).

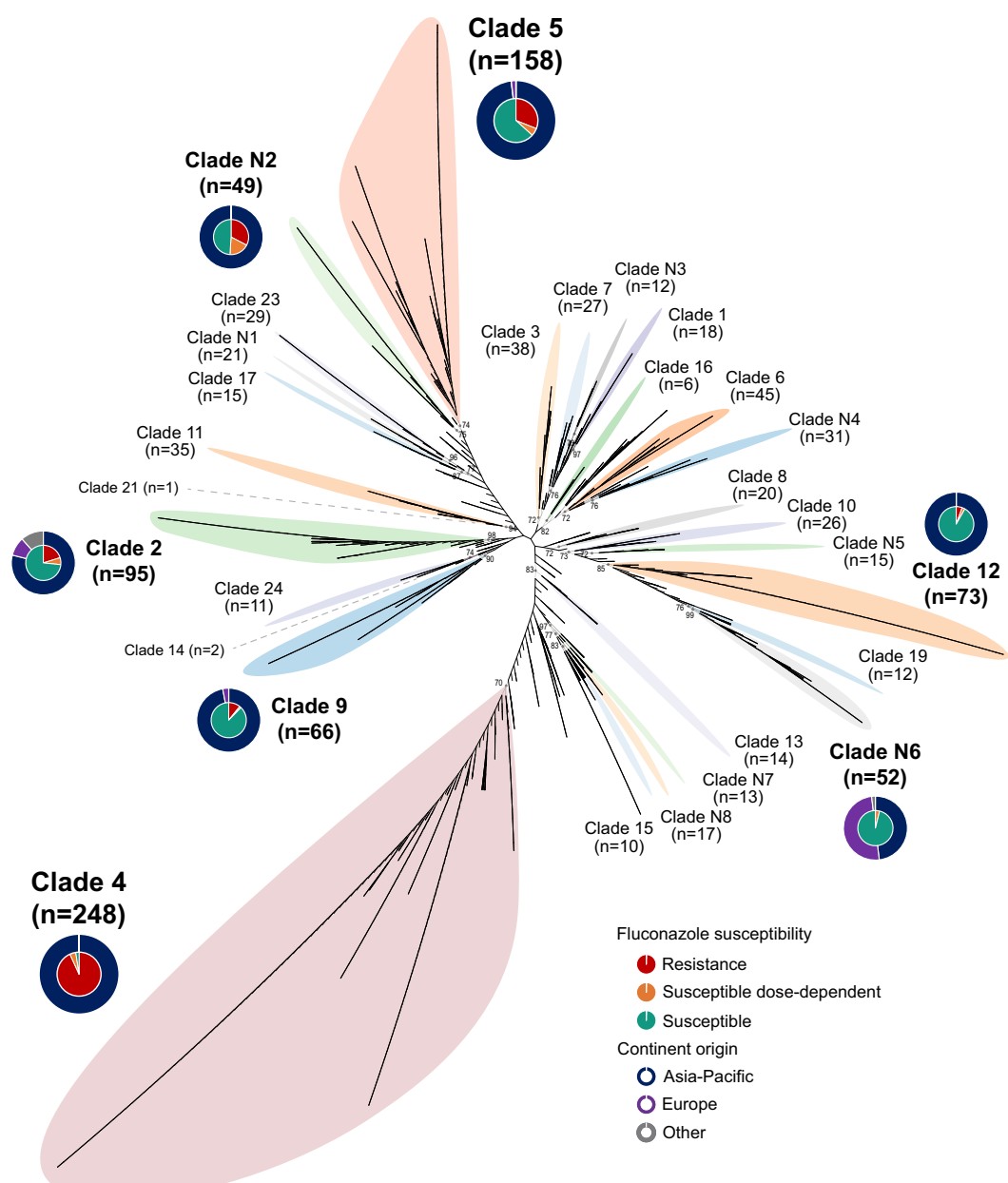

**Fig. 2 | Maximum-likelihood tree of 1571 *C. tropicalis* isolates based on concatenated sequences of six MLST gene loci.** An unrooted layout with an equal-angle method ignoring branch lengths was used. The major MLST clades identified were labeled by different background colors of phylogenetic tree branches. The pie charts represent antifungal susceptibilities (inner pie charts) and continental origins (outer ring charts) of major MLST clades (prevalence >3%).

was isolated in 1994 from North America (Supplementary Data 2). These long-term records provided an opportunity to analyze changes in major azole-resistant lineages over time (Fig. 3e). Of the two most prevalent azole-resistant genotypes, i.e., clades 4 and 5, the earliest fluconazole-resistant isolate in the latter clade were found in 1999 in Asia (Supplementary Data 2). During 1999–2008, clades 5 was the most common lineages among fluconazole-resistant isolates of the whole collection (Fig. 3e).

In comparison, the largest resistant clade, clade 4, emerged more recently, with the first isolate identified in 2010 within our collection in Asia (Supplementary Data 2). However, in a short period of time, the resistant population within clade 4 expanded quickly, and it became the predominant fluconazole-resistant genotype in 2009–2013 of the whole collection (52.7%), replacing the earlier-emerging clades 5, then continued to increase to 67.3% after 2014 (Fig. 3e). Moreover, clade 4 was the most common resistant lineage in

a vast area of Asia-Pacific, including in China (China's mainland and Taiwan, China), Thailand and Singapore from Asia and Australia from Oceania; the proportion of strains from clade 4 ranged from 34.7% to 86.7% among azole-resistant isolates in each region (Fig. 3d). These findings indicated that the emergence and wide dissemination of azole-resistant clade 4 isolates within last decades have become a unique challenge in Asia-Pacific.

**Population genomics reveal phylogenetic cluster AZR, an MLST clade 4 subpopulation that exhibits high-level azole resistance**
To gain a more in-depth understanding of the intra-population structure of MLST clade 4 isolates and potential correlations to antifungal resistance, we carried out a WGS-based analysis on 629 *C. tropicalis* strains. In general, 100% (458/458) of clinical isolates collected in China, along with 95.6% (173/181) of global strains achieved from previous studies reached ≥60× average read depth.

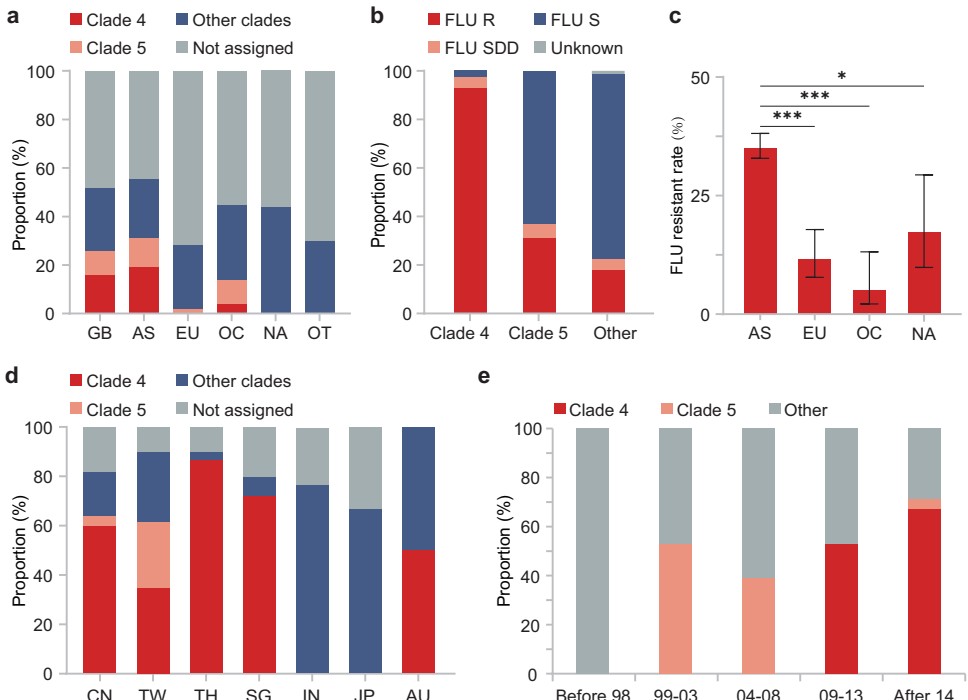

**Fig. 3 | Geographic variations of *C. tropicalis* MLST clades, their association with azole resistance, and related temporal trends. a** Geographic variations of MLST clades across different continents. GB global, AS Asia, EU Europe, OC Oceania, NA North America, OT other continents. **b** The overall fluconazole (FLU)-resistant (R), susceptible-dose-dependent (SDD) and susceptible (S) rates of clade 4 and clade 5 isolates vs strains from other phylogenetic populations. **c** Comparison of fluconazole-resistance rates across different continents. Resistance rates are presented by bar plots, and the error bars indicated 95% confidence intervals, with *p*-values determined using two-sided Fisher exact test with Bonferroni correction; *adjust *p*-values < 0.05; ***adjust *p*-values < 0.001. In all, as MLST clade 4 and clade 5

isolates were more prevalent in Asia, and these two clades exhibited notably decreased azole susceptibly, the azole-resistance rate was significantly higher in Asia than in North America (*p* = 0.016), Europe (*p* < 0.001) and Oceania (*p* < 0.001). **d** Proportion of clade 4 isolates among fluconazole-resistant isolates in Asia-Pacific regions. CN China's mainland, TW Taiwan China, TH Thailand, SG Singapore, IN India, JP Japan, AU Australia. **e** Proportion of clade 4 and clade 5 isolates among fluconazole-resistant strains over time. Clearly, clade 5 emerged earlier and became predominant from 1999 to 2008, but the major azole-resistant population quickly shifted to clade 4 after 2009.

Overall, 97.8% (615/629) of the *C. tropicalis* isolates studied were diploid (Fig. 4a). Polyploidies were rarely identified, with triploid, haploid and octoploid isolates only accounting for 1.6%, 0.3% and 0.3% of the collection, respectively (Fig. 4b–d). Aneuploidies were observed in 2.1% of strains (*n* = 13), and most aneuploidy events were found on chromosomes 5 (*n* = 5) or 6 (*n* = 5) (Fig. 4e–j, Supplementary Data 2). Additional large structural variations were observed on chromosome 4 in 4.9% (*n* = 31) of isolates, and on chromosomes 2, 3, 6 or R in a total of 17 isolates (Supplementary Data 2). Moreover, genome-wide high levels of heterozygosity were observed in 2.7% (17/629) of strains that reside in two separate phylogenetic branches (Supplementary Fig. S1, Supplementary Data 2). Of note, all aneuploidy isolates with extra complete chromosome 5 (*n* = 3) or isochromosome 5q (*n* = 1) that contained *ERG11* and *TAC1* gene loci, were resistant to fluconazole (Supplementary Data 2). However, antifungal resistance was not found to be associated with other polyploidy, aneuploidy, large structural variation or high heterozygosity genotype features.

By combining phylogenetic, average nucleotide identity (ANI) and ADMIXTURE population structure analyses of WGS data, a total of 36 *C. tropicalis* clusters (CTC) were proposed, and 22 of major WGS CTCs (number of isolates >5) were named (clusters CTC01-CTC20, cluster AZR, and one cluster nested in group AZR-ADJ) (Fig. 5, Supplementary Figs. S1 and S2). Strains assigned to these 22 WGS clusters accounted for 64.9% of the 629 isolates studied. Generally, the association between WGS phylogenetic clusters and MLST clades was revealed, but WGS exhibited higher discriminatory power. Of 22 genomic clusters, 21 only comprised strains from unique MLST clades (except for those isolates not assigned to any MLST clade) (Fig. 6a). The only exception was CTC08, which was made up of isolates from MLST clade

N7 (54.5%) and clade 19 (45.5%) that were located on two divergent branches in MLST maximum-likelihood tree (Figs. 2 and 6a).

The AZR cluster was the largest genomic cluster, accounting for 12.6% (79/629) of the collection (Figs. 5 and 6a). Within cluster AZR, strains' pairwise SNP differences varied from 2112 to 25,160 bp (median = 7731 bp), and no evidence of nosocomial outbreaks was observed. Of note, 97.5% (77/79) of AZR isolates were fluconazole-resistant (Fig. 6b), all of which were also cross-resistant to voriconazole. The remaining two isolates were susceptible dose-dependently to fluconazole (SDD) (Fig. 6b) and of intermediate susceptibility to voriconazole. In addition, 60.8% and 96.2% of AZR isolates were also of non-wild-type phenotype to itraconazole and posaconazole, respectively. Moreover, 92.4% of these isolates (73/79) exhibited high-level resistance to fluconazole, with minimum inhibitory concentrations (MICs) of at least 256 mg/L (Fig. 6c).

### A dramatic increase in azole-resistance rates correlated with population expansion in cluster AZR

Interestingly, we observed a branch of 28 isolates immediately adjacent to cluster AZR in the WGS phylogenetic tree that also exhibited a high rate of fluconazole non-susceptibility (92.9%, 26/28). This branch is referred to as "group AZR-ADJ" in this study (Figs. 5 and 6b). Of note, all isolates in cluster AZR and 96.4% (27/28) strains in the azole non-susceptible group AZR-ADJ carried well-recognized nucleotide mutations A395T/W and C461T/Y in their *ERG11* gene, which lead to amino acid substitutions Y132F and S154F in the azole antifungal target Erg11p (Fig. 6d). In addition, three AZR isolates (4.1%) carried an additional substitution, P397S (Fig. 6d). No fluconazole-susceptible isolates involved in this study carried any of these three substitutions. Of note,

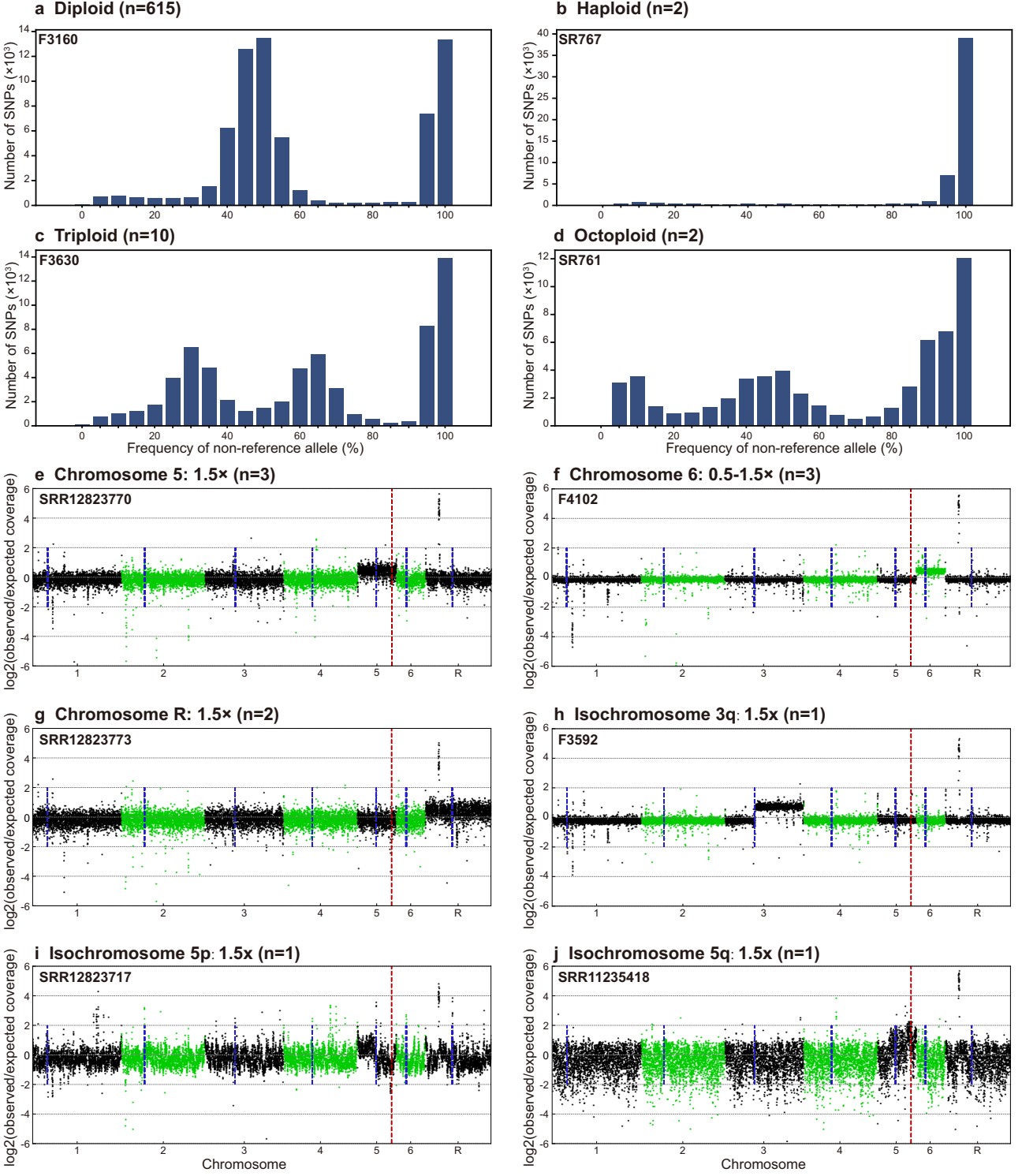

**Fig. 4 | Polyploidy and aneuploidy events observed in genomes of *C. tropicalis* isolates.** Results from representative strains are given for the different types of events, with strain ID no. labeled in the top-left corner of each figure.
**a**–**d** Polyploidy analysis based on the frequency of the non-reference allele for all heterozygous biallelic SNPs across the genome. The *x*-axis indicates the frequency of the non-reference allele, and the *y*-axis represents the cumulative number of SNPs. **a** Diploid isolates with peaks of allele frequency observed at 0.5. **b** Haploid isolates with peak of allele frequency not observed before 0.95. **c** Triploid isolates with peak of allele frequency found in the range of 0.33 and 0.66. **d** Octoploid isolates with peaks of allele frequency found at 0.12, 0.5 and 0.87. **e**–**j** Aneuploidy

events, which were confirmed by elevated coverage at the relevant locus (shown as dot plots: each dot represents a 1000-bp region, with green and black representing different chromosomes). The chromosome number is shown on the *x*-axis, and log₂(observed/expected coverage) is shown on the *y*-axis (where expected coverage is the average genome-wide coverage for a corresponding isolate). Coverage changes are given along with the number of isolates having corresponding events in the subtitles of each figure. Blue dash lines indicate centromere positions. Red dash lines indicate the position of the *ERG11* gene. **e** Chromosome 5 aneuploidy. **f** Chromosome 6 aneuploidy. **g** Chromosome R aneuploidy. **h** Isochromosome 3q. **i** Isochromosome 5p. **j** Isochromosome 5q.

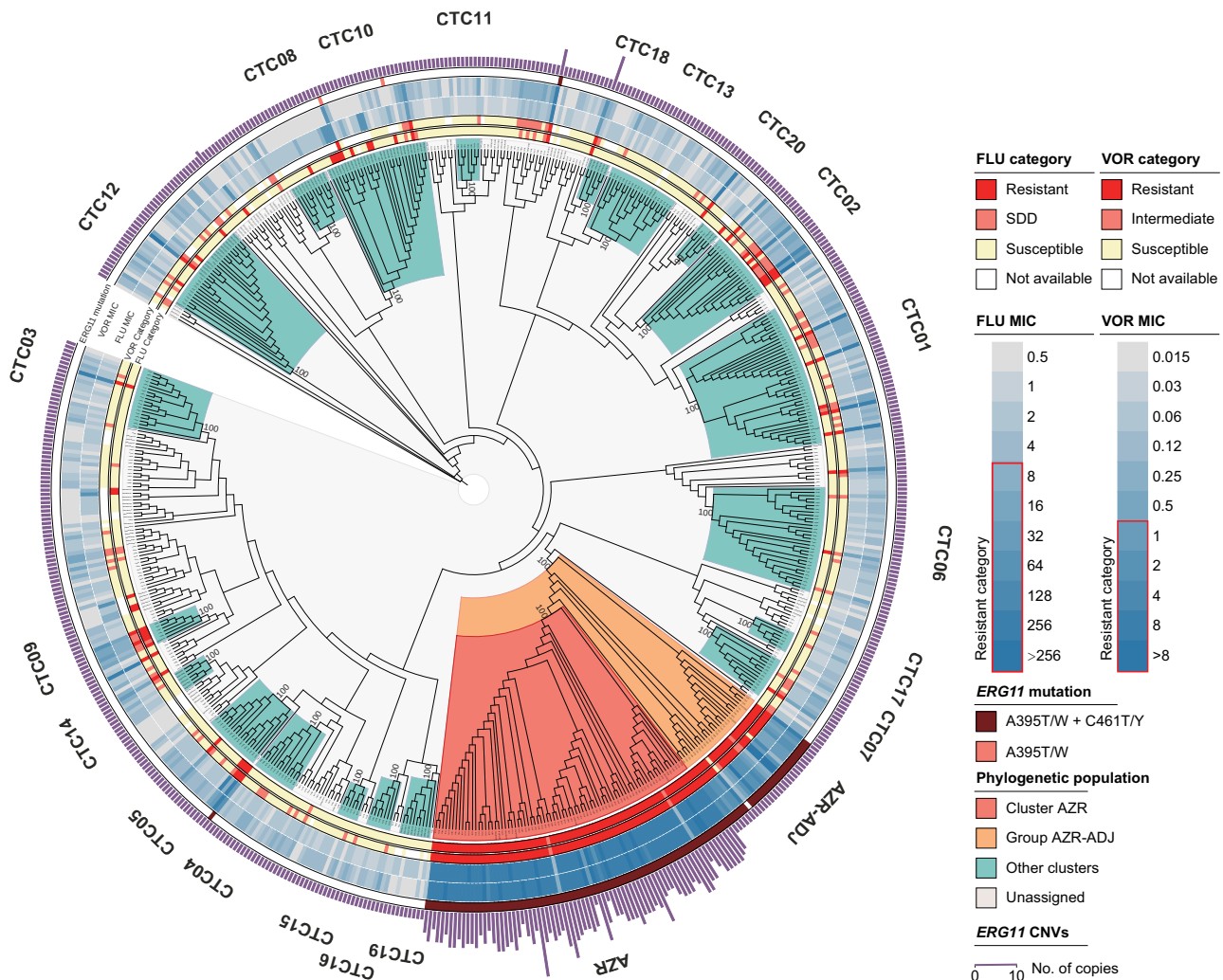

**Fig. 5 | A circular maximum-likelihood tree of 629 *C. tropicalis* isolates generated based on WGS SNPs.** Major phylogenetic clusters assigned–including cluster AZR, group AZR-ADJ, and CTCs 01-20–are labeled. In addition, the susceptibilities of strains to fluconazole (FLU) and voriconazole (VOR), and distributions of *ERG11* key mutations as well as *ERG11* copy number variations (CNVs), are indicated by different outer rings.

previous studies have confirmed that the mutations A395T/W, resulting in the amino acid substitution Y132F, play a crucial role in conferring azole resistance in *C. tropicalis*[14,15]. On the other hand, although mutation C461T/Y led to the substitution S154F is frequently observed together with mutations A395T/W, it does not independently contribute to the azole-resistant phenotype[15].

All the cluster AZR isolates belonged to MLST clade 4/CC1, but only 39.3% (11/28) of the AZR-ADJ isolates belonged to MLST clade 4, and none of AZR-ADJ isolates belonged to MLST CC1 ($p < 0.001$, Fig. 6d, e). The proportion of *ERG11*[A395T] homozygous mutant strains was higher in group AZR-ADJ (50.0%) than in cluster AZR (7.6%) ($p < 0.001$, Fig. 6f). Within group AZR-ADJ, *ERG11*[A395T] homozygous mutant strains exhibited significantly higher fluconazole distributions compared to heterozygous strains ($p = 0.002$). Most importantly, though high rates of fluconazole resistance were detected within both cluster AZR and group AZR-ADJ (97.5% vs 92.9%, $p = 0.33$, Fig. 6b), group AZR-ADJ isolates generally exhibited low- to moderate-levels of fluconazole resistance, as 80.8% of isolates had fluconazole MICs of at most 128 mg/L, and the fluconazole MIC distribution in group AZR-ADJ strains was significantly lower than that in cluster AZR isolates ($p < 0.001$).

Cluster AZR was found to be widely distributed over a vast geographic region. Though 98.7% (78/79) of isolates were found in Asia,

there was also one isolate detected in Oceania (Australia). Of 78 Asian cluster AZR isolates, 73 of them were identified in China and five were identified in Singapore, which accounted for 16.3% and 50.0% of isolates' collection in each country, respectively. Besides, 77.8% of 27 cities in China involved in this study have cluster AZR isolates detected (Fig. 1b). Moreover, a rapid expansion of cluster AZR isolates was observed in China over time, with its prevalence among fluconazole-resistant strains elevating from 0% in 2009 to 64% over 8 years ($p < 0.001$, Fig. 6g). In comparison, the proportion of group AZR-ADJ strains decreased from 100% to 4% over the same time period ($p < 0.001$, Fig. 6g). During this period, previous research showed that the rate of fluconazole resistance of *C. tropicalis* isolates collected in China dramatically increased from less than 6% to more than 30% (Fig. 6b)[8]. There was a significant association between the changing azole resistance and the change of major resistant clones ($p = 0.011$); we propose that the notable increase of azole resistance of *C. tropicalis* in China was led by the expansion of the population of isolates from cluster AZR.

## Detection of tandem gene duplication of *ERG11* gene in cluster AZR isolates

To explore potential mechanisms, in addition to *ERG11* gene mutations, that contribute to the distinctive high-level azole resistance

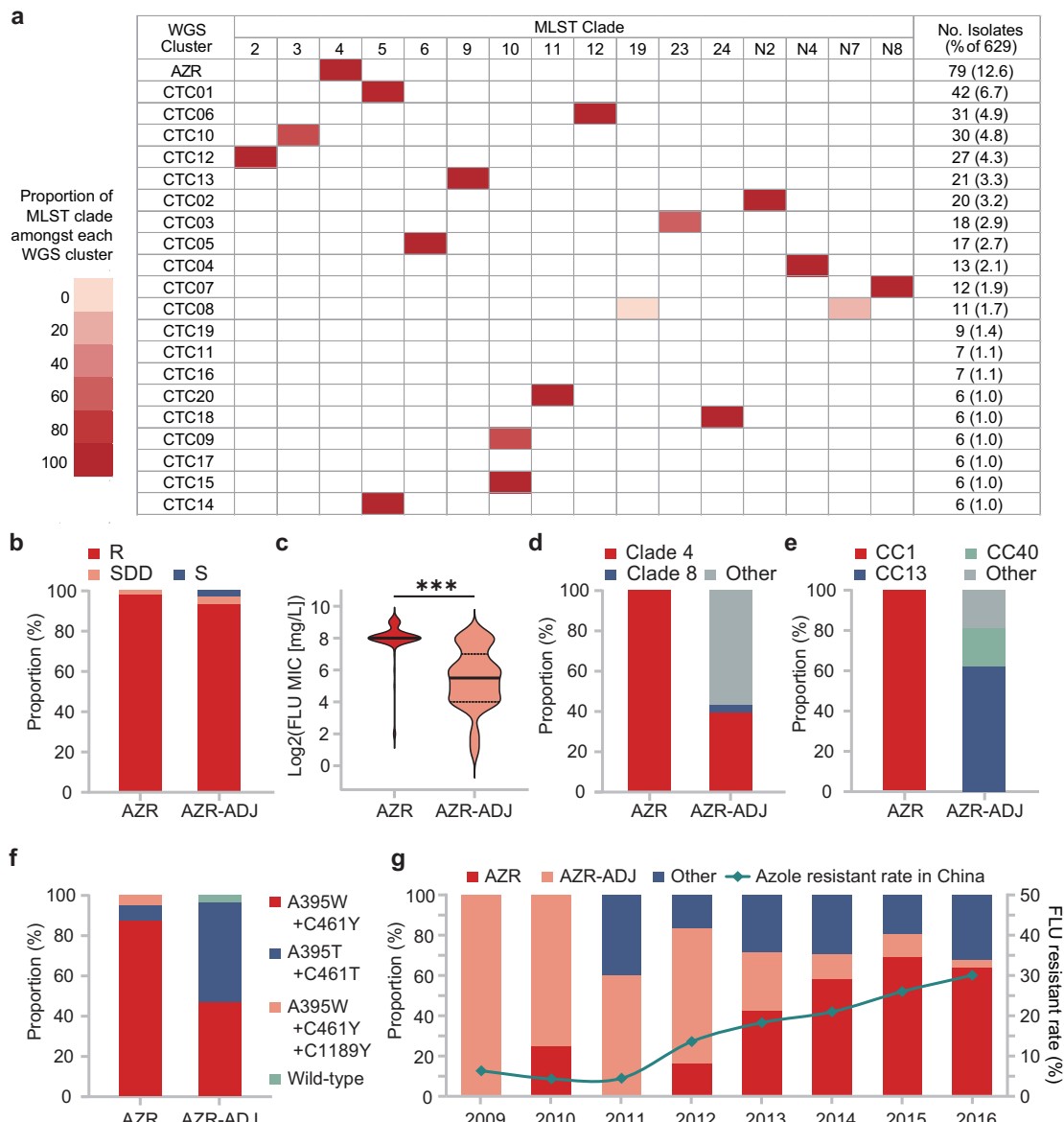

**Fig. 6 | Characterization of the associations of WGS phylogenetic cluster AZR and group AZR-ADJ with azole resistance and their changes over time.**
**a** Association between MLST clades and WGS clusters. **b** The overall fluconazole (FLU)-resistant (R), susceptible-dose-dependent (SDD) and susceptible (S) rates of cluster AZR and group AZR-ADJ isolates. Both groups had a resistance rate of >92.8%. **c** A comparison of the distribution of fluconazole MICs between cluster AZR and group AZR-ADJ. The majority of cluster AZR isolates were found to be highly resistant to fluconazole (MIC ≥ 256 mg/L), which is distinct from group AZR-

ADJ isolates, most of which exhibited low to moderate levels of azole resistance. ***p-values < 0.001. **d**, **e** Differences in the proportion of major MLST clades and clonal complexes (CCs) within the two populations, respectively. **f** Key nucleotide mutations observed in the *ERG11* gene carried by cluster AZR and group AZR-ADJ isolates. **g** The proportion of cluster AZR and group AZR-ADJ isolates among fluconazole-resistant strains over time, and the correlations of these trends with changes in the *C. tropicalis* fluconazole-resistance rate in China.

among cluster AZR isolates, we carried out comparative CNV analyses for 6160 open reading frames (ORFs) previously predicted in *C. tropicalis* genomes, using group AZR-ADJ strains as comparators. Of note, only a single gene in cluster AZR isolates exhibited an average sequencing depth elevation greater than 3 times (average depth 3.48 ± 1.01) vs group AZR-ADJ strains (average depth 1.06 ± 0.11), suggesting that notable CNV events occurred (*p* < 0.001, Fig. 7a). Functional annotation indicated that this ORF was the *ERG11* gene. There were no other ORFs in cluster AZR isolates observed to have elevations in sequencing depth of more than 2 times.

At the strain level, *ERG11* CNVs occurred in 97.4% (77/79) of cluster AZR isolates, but in none of group AZR-ADJ isolates. In addition, *ERG11* gene copy numbers varied from 3 to 16 copies among strains with CNVs (Fig. 7a, Supplementary Data 2).

Subsequent sequencing depth analyses of the *ERG11* gene and its adjacent 5′-upstream and 3′-downstream regions illustrated that the sizes of CNV segments and their breakpoint sequences were the same in all cluster AZR isolates. These segments started from −1164 bp in the 5′-upstream region of the *ERG11* gene, to 316 bp in the 3′-downstream region of the *ERG11* gene. Including the 1587 bp *ERG11* coding domain sequence, this region is composed of a 3067-bp segment (Fig. 7b).

To further characterize the spatial distribution of *ERG11* CNV segments in genomes of AZR strains, long-read sequencing was carried out on five selected AZR isolates and one AZR-ADJ isolate (Fig. 7c). All *ERG11* CNV segments appeared tandemly; that is, the *ERG11* CNVs resulted from tandem gene duplication events at the original *ERG11* gene position of the wild-type strain (Fig. 7e). Long-read sequencing

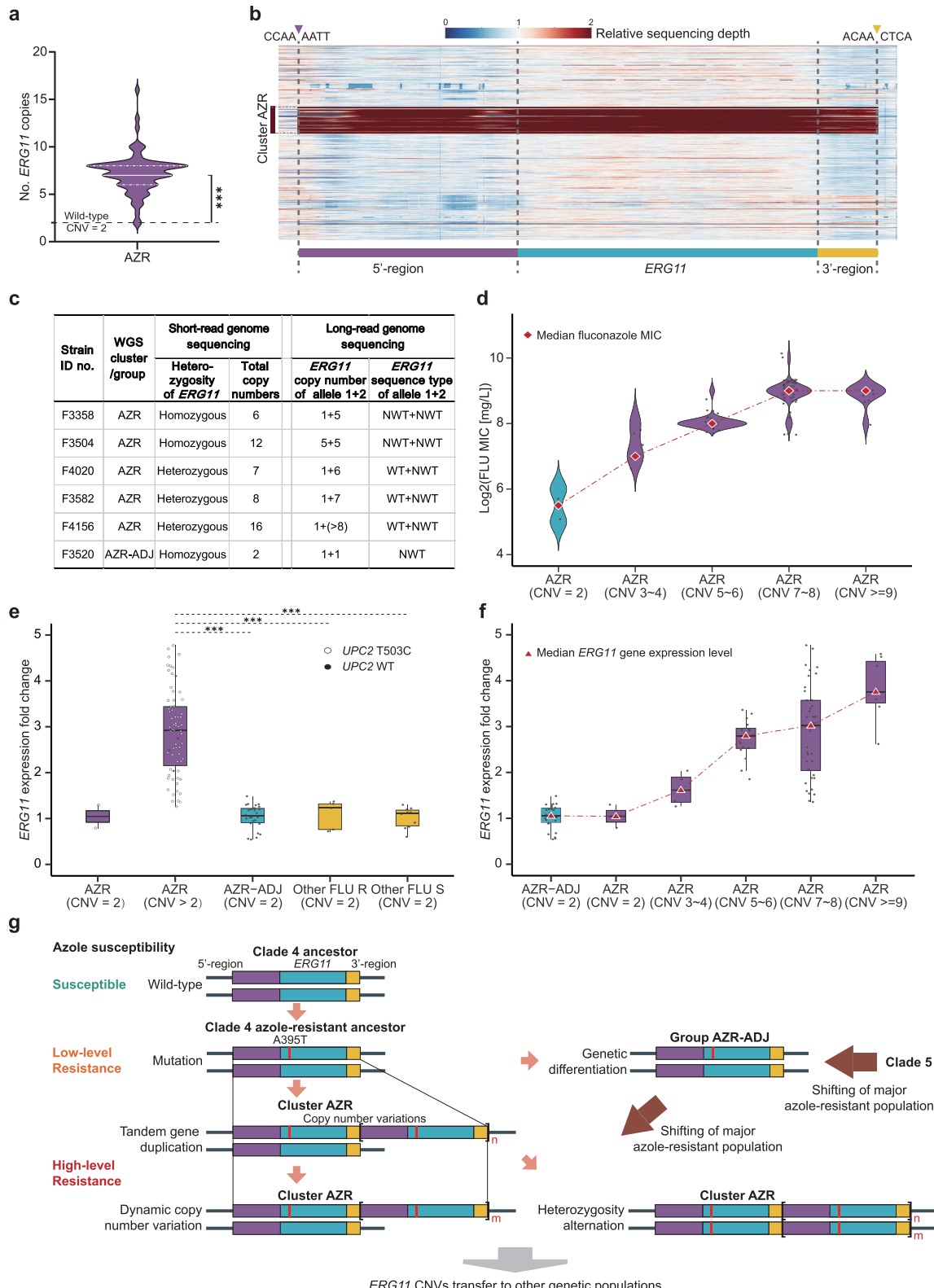

also supported the similarity of sizes of CNV segments and breakpoint sequences among all strains.

Moreover, short-read sequencing led to the finding that three of five AZR isolates had heterozygous *ERG11* gene alleles and the other two were homozygous. Long-read sequencing illustrated that tandem gene duplication events had only occurred on the chromosome with mutated *ERG11* alleles with the A395T and C461T mutations, while

chromosomes with the wild-type *ERG11* allele did not have gene duplications (Fig. 7c). In comparison, in two cluster AZR strains considered "homozygous" at the *ERG11* locus according to short-read sequencing, one strain (strain ID no. F3504) carried an equal copy number of *ERG11* genes ($n = 5$) on both alleles, while the other strain (strain ID no. F3358) carried a single copy of the mutant *ERG11* gene on one allele but five copies of mutant *ERG11* on the other allele (Fig. 7c).

**Fig. 7 | *ERG11* tandem gene duplications observed in *C. tropicalis* cluster AZR isolates. a** Distribution of *ERG11* CNVs observed in 76 cluster AZR isolates. ***p*-values < 0.001. **b** A heatmap of copy number variations (CNVs) of the *ERG11* gene observed among 629 isolations by short-read genome sequencing. Each row of the heatmap represents a single strain, and the order of strains from the top to the bottom is in accordance with what appears in Supplementary Data 1. Colors indicate the relative sequencing depth of each base pair within the region shown, using the average sequencing depth of each strain as a reference. The CNV segment identified in cluster AZR isolates comprised the complete 1587 bp *ERG11* coding domain sequence (blue), its 1164 bp 5′-upstream region (purple) and 316 bp 3′-downstream region (yellow). Up- and downstream break sequences shared by all strains with CNVs are labeled with purple and yellow triangles at the top. **c** Long-read sequencing of selected cluster AZR isolates indicated that *ERG11* CNVs resulted from tandem gene duplications. Moreover, CNVs occurred only on *ERG11* mutant alleles (with the A395T mutation). **d** Distribution of fluconazole (FLU) MICs to *ERG11* CNVs. It was observed that a rise in fluconazole MIC distribution was correlated with an increase in *ERG11* CNVs. **e** Comparison of *ERG11* gene expression levels among isolates in different phylogenetic populations. Box-and-whisker plots were used to present gene expression levels, and data points were overlayed as dot plots. The upper and lower whiskers represent maximum and minimum values, respectively; the hinges of the box represent 25% to 75% percentiles and the horizontal line within the box represents median values. Statistical analysis was carried out using two-sided Kruskal-Wallis H test and adjustments were made for multiple comparisons; ***adjusted *p*-values < 0.001. It was revealed that cluster AZR isolates with *ERG11* CNVs had significantly higher *ERG11* gene expression levels. **f** Association between *ERG11* gene expression levels and *ERG11* CNVs. It demonstrated that an increase in *ERG11* CNVs contributed to a significant upregulation of the gene expression. **g** A proposed evolution history of *C. tropicalis* clade 4/cluster AZR isolates.

Interestingly, among all clinical isolates not belonging to cluster AZR or group AZR-ADJ, an additional strain (ID no. F3784) was found to have the *ERG11* A395W mutation coupled with CNVs (Fig. 5). This strain also shared the sizes of CNV segments and breakpoint sequences with the cluster AZR strains.

## *ERG11* CNVs are associated with increases in azole resistance levels

Further investigations were conducted to determine if additional copies of *ERG11* were associated with an increase in azole resistance. By subgroup analysis of cluster AZR isolates that carried varying copy numbers of *ERG11* gene, it was observed that strains without *ERG11* CNV events, despite being resistant to fluconazole, had lower fluconazole MICs (32 to 64 mg/L) (Fig. 7d). In comparison, resistance azole resistance was more pronounced in isolates with *ERG11* CNVs, and as copy numbers of *ERG11* gene increased, a corresponding elevation in fluconazole MIC distributions was noticed. For isolates carrying 3–4 copies of the *ERG11* gene, the median fluconazole MIC value rose to 128 mg/L and isolates with 5–6 copies of *ERG11* gene displayed an even higher median fluconazole MIC value of 256 mg/L (Fig. 7d). The trend continued with cluster AZR isolates possessing 7–8 and ≥9 copies of *ERG11* gene, which exhibited the highest fluconazole MIC distributions with median fluconazole MIC values of 512 mg/L (Fig. 7d). These results suggested that the high-level azole resistance character of cluster AZR isolates were associated with the presence of *ERG11* CNVs.

Previous studies have shown that *ERG11* gene overexpression contributes to decreased azole susceptibility in multiple *Candida* species; such overexpression may result from increases in *ERG11* gene copy numbers, or it may be associated with mutations in transcription factors, such as *UPC2*, that regulate *ERG11* gene expression[22,24]. In this study, in addition to the *ERG11* CNVs identified, it was further revealed that 94.9% (75/79) of cluster AZR isolates had a missense mutation (T503C) in the *UPC2* gene (Supplementary Data 3).

To evaluate the potential contributions of *ERG11* CNVs and *UPC2* gene mutations to the expression level of the *ERG11* gene, qPCR was carried out within all cluster AZR isolates, using all group AZR-ADJ and a subset of other phylogenetic cluster (*n* = 15) strains as comparators and *C. tropicalis* strain ATCC750 as a reference (Fig. 7e, f). Generally, no significant changes in the level of *ERG11* gene expression were observed in group AZR-ADJ isolates, nor in two cluster AZR isolates that carried *UPC2* gene mutations but that did not carry *ERG11* CNVs (Fig. 7e). In comparison, an average increase of *ERG11* gene expression of 2.9-fold (IQR 2.2- to 3.6-fold) was observed among cluster AZR isolates with *ERG11* CNVs, including four strains that carried the wild-type *UPC2* gene (Fig. 7e). Furthermore, we observed a dose-dependent relationship between the copy numbers of *ERG11* gene and its expression level (Fig. 7f).

Missense mutations in other genes previously reported to be associated with azole resistance were also analyzed (Supplementary Data 3). Of note, some unique mutations were observed in drug efflux transporter-encoding genes and the genes that encode their transcription factors. For instance, 97.5% (77/79) of cluster AZR isolates had amino acid substitution A446E in Tac1 (Supplementary Data 3), a transcription factor controlling the expression of the Cdr drug efflux pump. In addition, 98.6% (78/79) of AZR and 82.1% (23/28) of AZR-ADJ isolates had amino acid substitution L129F in Stb5, a regulator of the Pdr drug efflux pump, and 92.2% (72/79) AZR isolates had an additional Stb5 substitution T313N (Supplementary Data 3). However, qPCR analysis revealed no significant changes in the expression level of *CDR* or *PDR* genes in cluster AZR isolates.

Overall, these findings strongly suggest an association between *ERG11* CNVs and the overexpression of the *ERG11* gene among cluster AZR isolates, indicating a potential mechanism contributing to the development of high-level azole-resistant phenotype within this population.

## Increase of *ERG11* gene copy numbers induced in vitro results in enhanced azole resistance

We further aimed to verify whether *ERG11* CNV plays an essential role in contributing to azole resistance. A cluster AZR strain (ID no. F4082, *ERG11* copy number = 3, fluconazole MIC = 128 mg/L) was serially passaged 10 times in vitro under fluconazole stress of 256 mg/L. The 0th, 3rd, 6th, and 10th passages were specifically selected to evaluate alterations in antifungal susceptibility, *ERG11* expression level, and *ERG11* CNVs (Fig. 8a). Overall, from the 0th to 6th passages, no notable changes were observed in fluconazole MICs, which remained constant at 128 mg/L (Fig. 8b). Correspondingly, short- and long-read sequencing confirmed that 3rd and 6th passages of the strain carried identical copy numbers of *ERG11* gene compared to the 0th passage (*n* = 3, Fig. 8c), and qPCR analysis indicated no variation in the expression level of the *ERG11* gene (Fig. 8d). However, a significant 2-fold increase in fluconazole MIC was observed in the 10th passage that reached 512 mg/L (Fig. 8c). This alteration coincided with an expansion of the *ERG11* gene copy number to *n* = 7 (Fig. 8b), and a corresponding 2-fold increase in the expression level of the *ERG11* gene compared to the 0th generation was observed (Fig. 8d).

Furthermore, to assess the stability of *ERG11* CNVs, we conducted an additional serially passaging of the original strain (0th passage) and its 10th passage for 10 times without subjecting it to fluconazole exposure. Our results showed no changes in both the fluconazole MICs and the copy numbers of the *ERG11* gene, which suggests that once *ERG11* CNVs have emerged, it has the potential to maintain stability and persist in the absence of external azole stresses. Consequently, this in vitro experiment strengthened the evidence that *ERG11* CNVs play a contributory role in the development of high-level azole resistance in *C. tropicalis*.

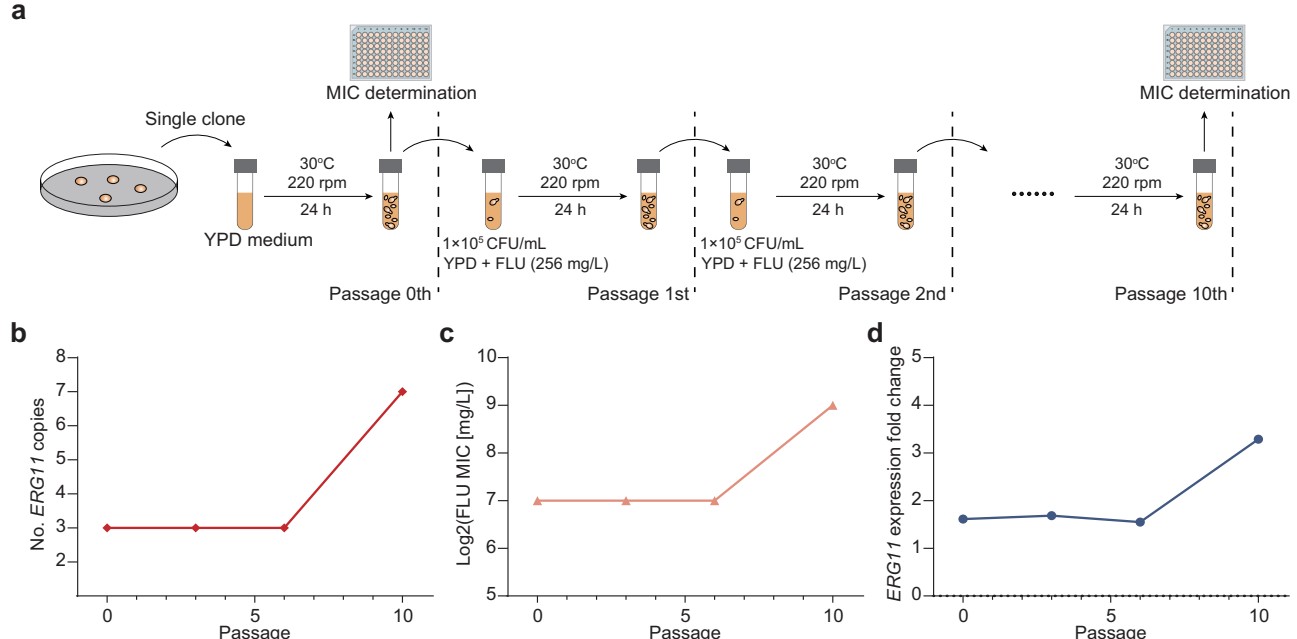

**Fig. 8 | ERG11 gene copy number increasing induced in vitro results in enhanced azole resistance. a** Design of the in vitro induction experiment. Cluster AZR strain ID no. F4082 (*ERG11* copy number = 3, fluconazole MIC = 128 mg/L) was selected and passaged 10 times in vitro under fluconazole stress of 256 mg/L. The 0th, 3rd, 6th, and 10th passages were specifically selected to evaluate alterations in antifungal susceptibility, *ERG11* expression level, and *ERG11* CNVs. **b** At the 10th passage, the *ERG11* gene copy number increased to $n = 7$. **c** A 2-fold increase in fluconazole MIC was observed at the 10th passage. **d** Expression level of the *ERG11* gene also showed a 2-fold increase; $n = 3$ independent experiments were carried out, and data was presented as mean values ± SD.

## Ty3/gypsy-like retrotransposons are enriched in cluster AZR isolates

To further identify unique genomic features of cluster AZR isolates, a pan-genome analysis was carried out. A total of 16,824 clusters of gene orthologous groups (OGs) were identified, and a core genome composed of 4606 genes was found (Fig. 9a). A pan-genome-based comparative analysis only identified two OGs that were carried by a large majority (>90%) of cluster AZR isolates but that were less commonly seen (<20%) in other isolates, including strains of group AZR-ADJ (Supplementary Data 4). Of these two OGs, OG06397 was >1000 bp in length and is considered to be a functional gene. Homology-based annotation indicated that the product of OG06397 is a gag-pol fusion protein and that this gene is a *Ty3/gypsy*-like retrotransposon (Supplementary Data 4). When pooling cluster AZR and group AZR-ADJ isolates together and comparing them with other strains, two additional OGs were identified, both of which were predicted to be putative transposable elements, and one of the two (OG06243) was also a *Ty3/gypsy*-like retrotransposon (Supplementary Data 4).

To verify whether *Ty3/gypsy*-like retrotransposons were more enriched in azole-resistant populations, we further screened all *Ty3/gypsy*-like retrotransposons among all OGs, and five homologous OGs were identified (Fig. 9b). Of note, the proportions of isolates carrying each of these five OGs were all significantly higher in cluster AZR isolates than in strains belonging to other phylogenetic populations (Fig. 9b). Moreover, the overall cumulative copy number of retrotransposons was also significantly higher in cluster AZR isolates (median 14, IQR 12–15) than in group AZR-ADJ isolates (median 11, IQR 7–13, $p < 0.001$) and in other population strains (median 5, IQR 3–8, $p < 0.001$) (Fig. 9c). Along with the *ERG11* CNVs that were observed, the enrichment of *Ty3/gypsy* retrotransposons in cluster AZR isolates are another indicator of higher genomic plasticity of cluster AZR.

## Discussion

To date, the human impact of fungal diseases remains largely underrecognized, even though fungi are estimated to affect billions of people and cause 1.5 million deaths per year[2,25]. Invasive fungal diseases are associated with notably high mortality (20–50%), despite the availability of antifungal treatments[3–5]. Moreover, it is generally recognized that the prevalence of invasive fungal diseases is rising, because of the continuing expansion of high-risk populations, such as immunocompromised individuals, those receiving surgical operations or other invasive medical interventions, or patients in long-term intensive care units[5]. Invasive fungal diseases are associated with prolonged hospitalization and extra financial costs (more than €35,000/patient).

Compounding the problem of fungal diseases is the emergence of antifungal resistance as an important clinical problem[5,14]. To date, only a limited number of primary classes of antifungal agents, i.e., azoles, polyenes, flucytosine, and echinocandins, are available for treating invasive fungal diseases, and the loss of any of these options due to resistance will pose great challenges in patient management[14]. Among all antifungals, fluconazole and other azoles have been widely used as first-line therapies since the early 1990s, but increasing resistance to azoles quickly raised public concerns, and resistance has already driven changes to clinical practice[26,27].

It has been noted that epidemiology and antifungal resistance related to invasive candidiasis have significant geographic variations[3,6,24]. For instance, in North America, increasing azole resistance was more notable in *C. glabrata*[6,24,26], while challenges posed by azole-resistant *C. tropicalis* were more serious in the Asia-Pacific region[7,8,24]. In Australia, the rate of fluconazole resistance increased from 2% in the mid-2000s to greater than 16% in a decade; meanwhile, the rate of azole resistance increased more dramatically in China: from 6% in 2009 to more than 30% within ten years[7,8].

Of note, rapid or broad dissemination of antimicrobial resistance in certain species could be attributed to the emergence of specific phylogenetic populations, a process that has been recognized in varieties of medically important microbes, such as the prokaryotic pathogens *Mycobacterium tuberculosis*[28], *Salmonella typhi*[29] and *Clostridium difficile*[30], as well as within the most prevalent fungal pathogens

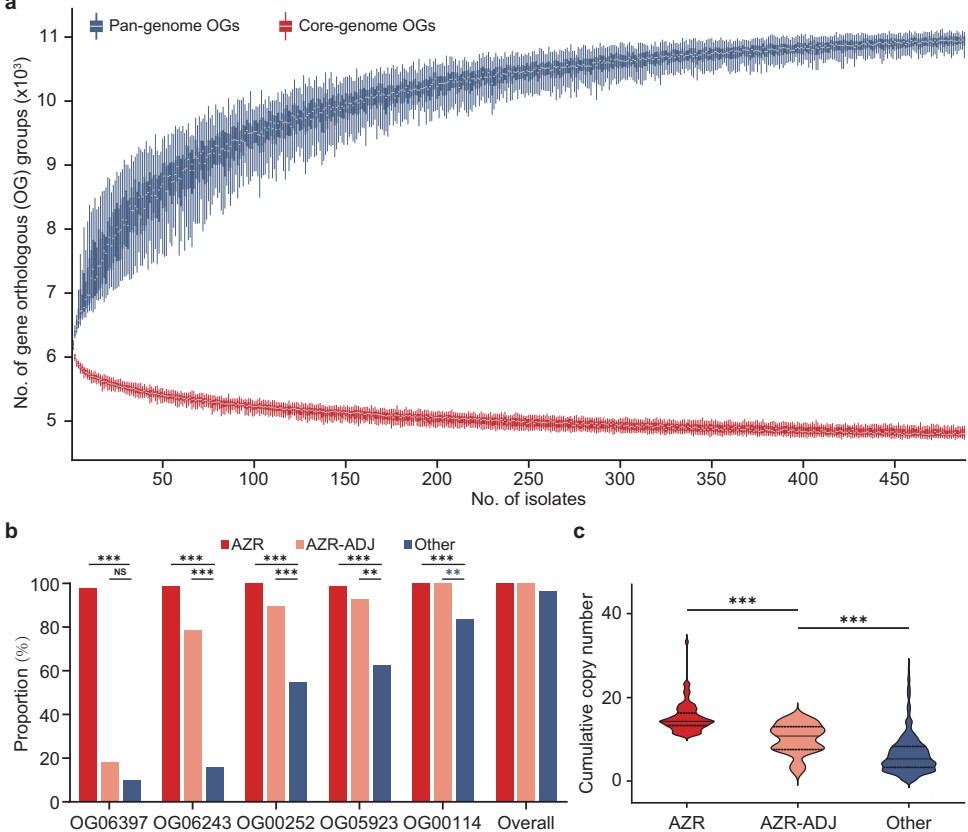

**Fig. 9 | Characterization of *Ty3/gypsy*-like retrotransposons enriched in cluster AZR isolates. a** Characteristics of the *C. tropicalis* pan-genome and core genome. The *y*-axis indicates the sizes of the pan-genome and core genome (with the number of predicted gene orthologous groups [OGs]), and the *x*-axis is the number of *C. tropicalis* strains incorporated for analysis. Box-and-whisker plots were used, with upper and lower whiskers represent maximum and minimum values, respectively. The hinges of the box represent 25% to 75% percentiles and the horizontal line within the box represents median values. **b** Comparison of

proportions of isolates carrying different *Ty3/gypsy*-like retrotransposons among cluster AZR, group AZR-ADJ and other phylogenetic populations. NS, not significant; **$p$-values < 0.01. ***$p$-values < 0.001. **c** Comparison of the cumulative number of *Ty3/gypsy*-like retrotransposons in different phylogenetic populations (shown as a bar-plot with median and interquartile range), which indicates that *Ty3/gypsy*-like retrotransposons are significantly more enriched in cluster AZR isolates. ***$p$-values < 0.001.

*C. albicans* and *Aspergillus fumigatus*[10,31]. Moreover, the expansion of resistant populations occurs not only among human individuals but also in broader contexts, e.g., from animals or the environment to humans[31,32]. For instance, there has been compelling phylogenomic evidence supporting that the acquisition of azole resistance in *A. fumigatus* is due to the widespread usage of azole agents in agriculture[31,33].

In *C. tropicalis*, the MLST scheme has been used for illustrating the phylogenetic relatedness of strains collected globally and has provided longitudinal insights into trends of clonal changes[13,34]. Using MLST, a study in Taiwan indicated that specific *C. tropicalis* clones associated with azole resistance had emerged earlier than 2000[34], while a shift of predominant resistant clones from MLST clade 5 to clade 4 was noted around 2010[13]. We also observed that in 1571 strains analyzed by MLST, clade 4 azole-resistant isolates were more prevalent (15.8%) than those of clade 5 (10.1%), and azole resistance of *C. tropicalis* in broad Asia-Pacific regions, such as in China, Singapore, Thailand, and Australia, were significantly associated with clade 4[7,13,35]. Furthermore, azole-resistant isolates of *C. tropicalis* belonged to MLST clade 4 (e.g., DST225) and clade 5 (e.g., DST140) have also been identified in environmental sources such as soil and fruits, which suggests the potential spread of resistant strains between the human population and the environment[36,37].

However, due to the diploid nature of *C. tropicalis*, Sanger sequencing-based MLST analysis requires manual checking for

heterozygosity at each base, which is labor-consuming and can potentially lead to the unintentional oversight of certain heterozygous sites[38]. One strength of the present study is that WGS was applied for a more precise characterization of *C. tropicalis* population structure in a large collection of strains. It has been recognized that mutations in the *ERG11* gene, particularly A395T, which results in the amino acid substitution Y132F, are one of the major mechanisms that contribute to azole resistance of clade 4 isolates[7,13,15,35]. WGS analysis in this study revealed that 71.8% (79/110) of isolates carrying the *ERG11*[A395T] mutation belong to a specific subpopulation of *C. tropicalis* MLST clade 4, which was named cluster AZR. In addition, 22.7% (25/110) of isolates with the A395T mutation were assigned to group AZR-ADJ, which is phylogenetically adjacent to cluster AZR. Of note, though both cluster AZR and group AZR-ADJ isolates had high rates of azole resistance (rates of fluconazole of 97.5% and 92.9%, respectively), the azole MIC of cluster AZR isolates was significantly higher than that of group AZR-ADJ isolates. More importantly, it was found that the dramatic increase of azole resistance in China was associated with the rapid population expansion of cluster AZR isolates, while the prevalence of AZR-ADJ isolates decreased (Fig. 6g).

Gene expression analysis of clinical isolates collected in this study further revealed that overexpression of the *ERG11* gene was significantly associated with high-level azole resistance phenotypes, and it was associated with MIC discrepancies between cluster AZR and group AZR-ADJ isolates. This overexpression may have also resulted in

shifts in the prevalence of the clones. It has been noted that *ERG11* overexpression contributes to azole resistance[15,24,39], and resistance mechanisms known to occur in *C. tropicalis* had been limited to sequence mutations in or upregulation of expression of *UPC2*, a gene encoding a transcription factor that regulates *ERG11* expression[7,16]. In this study, we also identified a mutation leading to the L168P substitution in the *UPC2* gene of 94.9% of cluster AZR isolates but not in any of group AZR-ADJ isolates. This mutation had been previously reported in clade 4 azole-resistant isolates in Australia[7]; however, no correlation between this *UPC2* mutation and the level of *ERG11* expression was found by qPCR.

Previous studies in different yeast species have illustrated that duplication of chromosome segments containing key antifungal-resistant genes may contribute to gene overexpression and result in resistance[14,21]. For instance, an isochromosome composed of duplicated left arms of chromosome 5 in *C. albicans*, which led to CNVs of *ERG11* and *TAC1* located in this region, is known to cause resistance to azole agents[21,22]. More recently, several studies have illustrated that in the emerging multi-drug-resistant species of global concern, *C. auris*, there are also strains with *ERG11* CNVs that are associated with increased gene expression and consequently high-level azole resistance[40,41]. In *C. tropicalis*, there was only one previous report that observed increased *ERG11* gene copy number; this increase was induced in vivo during antifungal therapy in a facility in Canada (also involved in WGS analysis of this study, strain ID no. SRR11235418, Supplementary Data 1)[23]. This particular strain had a similar resistant mechanism to that described above in *C. albicans*, which carried an additional isochromosome 5q that contains *ERG11* and *TAC1* genes. It did not carry any missense mutations in the *ERG11* gene and only exhibited a moderate level of azole resistance (fluconazole MIC of 32 mg/L). Furthermore, we identified two additional isolates from previous reports (strain ID no. SRR12823717 and SRR11235418, Supplementary Data 1), that exhibited aneuploidy involving an entire chromosome 5 and were resistant to fluconazole. Nevertheless, the prevalence of azole resistance induced by chromosome-level aneuploidy in *C. tropicalis* remained rare.

In comparison, our WGS analysis uncovered a notable prevalence of *ERG11* CNVs occurred among 78 of 629 isolates (12.4%) in this study, which did not arise from chromosome aneuploidy but rather were characterized by the duplication of a specific 1587 bp-chromosome segment containing the *ERG11* gene-coding region. Specifically, 77 (98.7%) of these isolates belonged to cluster AZR of MLST clade 4. More importantly, all *ERG11* alleles with CNVs were non-wild-type and had the A395T mutation. A combination of short- and long-read WGS sequencing further allowed for a better understanding of CNV segment characteristics. We found that *ERG11* CNVs occurring in cluster AZR isolates resulted from tandem gene duplications[17,19], as the 1587 bp-chromosome segments, which comprised the complete *ERG11* coding domain sequences and its 5′-upstream (1164 bp) and 3′-downstream (316 bp) regions, appeared recurrently in isolates' genomes. In addition, all segments shared breakpoint sequences. To the best of our knowledge, this is the first report for the occurrence of CNVs in the *ERG11* gene that were observed in a tandem-repeat form and were accompanied by important azole-resistant mutations.

Notably, a clear dose-dependent relationship was observed between the copy numbers of the *ERG11* gene, the gene's expression level, and the distribution of fluconazole MICs among cluster AZR isolates. To further validate the role of the *ERG11* CNV mechanism in azole resistance, an in vitro experiment was conducted. A cluster AZR isolate with a low copy number of the *ERG11* gene ($n = 3$, fluconazole MIC = 128 mg/L) was subjected to azole stress conditions and passaged 10 times. Remarkably, the copy number of the *ERG11* gene in the 10th passage expanded to $n = 7$. Concurrently, this increase was accompanied by a 2-fold elevation in the expression level of the *ERG11* gene, and a corresponding 2-fold rise in the fluconazole MIC value (which

reached 512 mg/L) compared to the 0th passage. Collectively, these findings provide enhanced evidence that the *ERG11* CNV mechanism contributes to the high-level fluconazole resistance observed in cluster AZR isolates relative to other *C. tropicalis* populations.

Previous studies have shown that mechanisms of CNV formation are quite similar among different organisms, even across prokaryotic and eukaryotic kingdoms[17]. Tandem gene duplications with recurrent end-points, like the *ERG11* CNVs observed in cluster AZR isolates in this study, generally arise by homologous recombination between repeated sequences, in a process called non-allelic homologous recombination (NAHR)[17,21]. NAHR can occur in double-strand break-induced recombination by double Holliday junction double-strand break repair, or when collapsed or broken replication forks are repaired by break-induced replication, which can produce changes in copy numbers[17,42]. In addition to NAHR, microhomology-induced replications and unequal crossovers have also been characterized as mechanisms responsible for gene duplications and deletions[17,42]. As shown in Fig. 7e, our data indicate that the common ancestor of cluster AZR and group ADJ isolates acquired *ERG11* mutation A395T and gained azole resistance first, then *ERG11* CNV events occurred in the cluster AZR population. These CNV events would further enhance the fitness of cluster AZR isolates, particularly toward antifungal stresses[14,20,21]. Thereafter, a rapid expansion of the cluster AZR population occurred, allowing these organisms to surpass other resistant phylogenetic groups, and introducing a dramatic increase of azole resistance in China.

It has been recognized that transposable elements in genomes have contributed significantly to genomic plasticity[14,43,44]. In this study, pan-genome analyses revealed four putative functional gene OGs, all of which were transposable elements, that were enriched in cluster AZR and/or in group AZR-ADJ isolates but were rarely identified in other phylogenetic populations. Moreover, two of the four OGs were *Ty3/gypsy*-like retrotransposons, and the cumulative copy numbers of all *Ty3/gypsy* homologies were also significantly higher in cluster AZR isolates. The *Ty3/gypsy*-like retrotransposon is one of the class-I transposable elements with long terminal repeats (LTRs) in yeasts, and they have also been detected in wide ranges of eukaryotes[43,45]. Previous studies have demonstrated that retrotransposons bearing LTRs in yeasts are associated with chromosome segment CNVs, LOHs, and large aneuploidy events[14,43,44]. Here, our data suggest that the enrichment of *Ty3/gypsy*-like retrotransposons in cluster AZR may be another mechanism that is associated with enhanced genomic variability and contributed to the rapid expansion of this population.

We acknowledge some limitations of the study. First, regarding sample inclusion, there is a noticeable geographic imbalance. While our MLST analysis included *C. tropicalis* strains from five continents, isolates from Asia accounted for over 80% of the total. This bias is attributed to a significant number of relevant studies published in Asia. Furthermore, given the higher prevalence of azole-resistant *C. tropicalis* in the Asian region, and many studies (including this one) primarily focused on the molecular epidemiology of resistant population, there is an overrepresentation of resistant strains in our analysis. Second, in our investigation of the molecular mechanisms, while our subgroup analysis of cluster AZR clinical strains and in vitro-induced azole resistance experiments demonstrated a dose-dependent correlation between *ERG11* CNVs and azole MICs, the role of the CNV mechanism was not solidly verified by reverting the tandem duplicated *ERG11* genotype back to a single-copy allele to examine changes in azole susceptibility. Additionally, the regulatory mechanism of *ERG11* CNV events remains not well-illustrated. Besides, two cluster AZR strains carrying *ERG11*[A395T] CNVs that did not exhibit high-level resistance to fluconazole (categorized as SDD) were found, which suggests the presence of additional mechanisms that may affect the susceptibility phenotype of these strains. Third, there's a lack of a standardized methodology for WGS bioinformatics in fungi, which can potentially

lead to discrepancies in interpreting SNPs and heterozygosity. Moreover, interpreting heterozygosity in genes with CNVs may have a reduced level of confidence. Nevertheless, these limitations do not detract from the substantial implications of our findings regarding the escalating antifungal resistance threats posed by the rapid expansion of the cluster AZR population. To implement effective antifungal stewardship strategies, our results advocate for the targeted monitoring of the cluster AZR population, which can serve as a valuable indicator to assess the severity of antifungal resistance challenges over time and evaluate the effectiveness of transmission control measures[24,33]. Furthermore, our in vitro experiment has demonstrated that cluster AZR can acquire a stronger resistant phenotype under azole stress through the *ERG* CNV mechanism. This finding once again emphasizes the critical importance of controlling irrational use of antifungal agents both in clinical settings and the environment[24,33].

In conclusion, a large-scale global phylogenetic analysis of *C. tropicalis* isolates revealed that MLST clade 4 has replaced other early clades and become the predominant azole-resistant population in Asia-Pacific regions, and this population expansion has led to high azole-resistant rates in these regions. Moreover, WGS approaches identified the emergence and rapid spread of cluster AZR, a sublineage of clade 4, which is responsible for the dramatic increase of azole resistance in China over the last several decades. Cluster AZR isolates exhibit extremely high-level azole resistance phenotypes. The tandem gene duplications of the *ERG11* gene, coupled with the key mutation A395T, emerged as a distinctive characteristic observed in cluster AZR isolates. This specific CNV mechanism is strongly associated with an elevation in *ERG11* gene expression levels and an increase in fluconazole MIC of the strains. Moreover, the enrichment of *Ty3/gypsy*-like retrotransposons in cluster AZR isolates may have further contributed to higher genomic plasticity in response to environmental stresses. Maintaining ongoing epidemiological surveillance of the clade 4/cluster AZR population is imperative, and there is an urgent need to develop effective strategies to address the growing threat of antifungal resistance posed by *C. tropicalis*.

## Methods

### *C. tropicalis* isolates

In the present study, we analyzed whole-genome data of 629 *C. tropicalis* strains, including all available genomes that were released in the NCBI database (https://www.ncbi.nlm.nih.gov/sra) and described in previous publications until Dec 31, 2022 (181 international strains from nine previous studies in all, Supplementary Data 2). In addition, 448 isolates causing human IFDs were collected from 49 hospitals in 27 cities in 27 provinces in China in a CHIF-NET surveillance program (Fig. 1b, Supplementary Data 2)[8]. Of note, to gain a more comprehensive understanding of potential longitudinal changes in the azole-resistant population, we ensured a similar proportion of azole-resistant strains (30% ± 5%) within each surveillance year, and strains from each susceptibility category were randomly selected from the CHIF-NET study.

Furthermore, all available *C. tropicalis* MLST data until Dec 31, 2022, were acquired from published literature and a public database (http://www.pubmlst.net). Redundant data from the same isolates were eliminated manually, and isolates without azole susceptibility results reported were also excluded, making an additional dataset comprising 942 isolates. Therefore, a total of 1571 *C. tropicalis* strains were used in this study (listed in Supplementary Data 1).

Generally, 80.8% of the isolates in this dataset were from Asia, as previous *C. tropicalis* MLST studies carried out were mainly performed in Asia, but the dataset also included isolates from Europe (10.2%), Oceania (4.7%), North America (3.6%), Latin America (0.4%) (Fig. 1a, c). In addition, of 1481 isolates with known isolation sources, 97.9% (*n* = 1450) were clinical strains causing human infections (Fig. 1c). Out of the 1376 clinical isolates with detailed information on isolation

sources, 58.1% (*n* = 782) were obtained from bloodstream infections. In comparison, a few (*n* = 28, 1.9%) isolates were from environmental sources (Fig. 1c).

### Genome resequencing and quality filtering

All 448 fungal isolates collected in China were grown on Sabouraud Dextrose Agar (SDA, Oxoid, Thermo Fisher Scientific: CM0041T) at 28 °C for 48 h, and subjected to short-read sequencing. DNA extraction was performed using a QIAamp DNA Mini Kit (Qiagen: 51306). A paired-end library with an average insert size of 300 bp was prepared with a Nextera XT DNA Library Preparation Kit (Illumina: FC-131-1096) and sequenced using the Illumina HiSeq X10 platform (Illumina, California, USA) in PE150 (paired-end sequencing, 150 bp reads) sequencing mode. Raw FASTQ files were processed through a standard pipeline through Trimmomatic (version 0.36) to remove the adapters[46], bases with phred score less than 3 and N bases in the leading and trailing parts, and bases with phred score less than 20 and length less than 36 in the 4 bp window.

In addition, long-read sequencing was carried out to characterize the spatial distribution of *ERG11* CNV segments in genomes. Five selected cluster AZR isolates and one AZR-ADJ isolate were tested (Fig. 7c). Libraries were prepared according to the manufacturer's instructions for the PacBio Template Prep Kit (Pacific Biosciences: 100-938-900) using a 10 kb-template preparation protocol, then sequencing was carried out on the single-molecule real-time (SMRT) platform (PacBio Biosciences, Menlo Park, USA) at Novogene Co. Ltd. (Tianjin, China). Low-quality reads were filtered by SMRT 2.3.0 software.

### Read mapping, variant identification and filtering

For each isolate, the filtered reads were mapped to the *C. tropicalis* MYA-3404 reference genome (GenBank assembly accession: GCA_013177555.1 [https://www.ncbi.nlm.nih.gov/datasets/genome/GCA_013177555.1/]) with the Burrows–Wheeler Aligner (BWA version 0.7.7) using the BWA-MEM algorithm[47]. The "view" command of SAMtools (version 1.6)[48] with the "-b" argument was used to generate the BAM file, and then the "sort "command was used to sort the BAM file with the default argument. Indel and variant calling were performed with the Genome Analysis Toolkit (GATK) (version 4.3.0.0)[49] according to the GATK Best Practices[50]. For each sample, variants were first marked by GATK MarkDuplicates, then called with GATK HaplotypeCaller to create single-sample GVCF files with the option "−emitRefConfidence GVCF" and using GATK CombineGVCFs to aggregate the GVCF files, using the default parameters including: −sample-ploidy = 2; −heterozygosity = 0.001 (for computing prior likelihoods). The GVCF files were then jointly genotyped with GATK GenotypeGVCF to produce a single multiple-sample SNP file containing data on every strain. Finally, SNPs were selected using GATK SelectVariants with the option "-select-type SNP" and filtered using the following parameters: VariantFiltration, QD < 2.0; ReadPosRankSum <−8.0; FS > 60.0; MQRankSum <−12.5; MQ < 40.0; and HaplotypeScore >13.0. INDELs were selected using GATK Select-Variants with the option "-select-type INDEL" and filtered using the following parameters: QD < 2.0, QUAL < 30.0, FS > 200.0 and ReadPosRankSum < -20.0. A "bamdst" script (https://github.com/shiquan/bamdst) was used for quality control by analyzing the BAM files, and samples with a low fraction of reads mapped (<60%) were excluded for further analysis. In addition, since *C. tropicalis* is a diploid species, an average read depth of 60× was considered to provide a high level of confidence for subsequent analyses.

### Multilocus sequence typing (MLST) analysis

MLST analysis was conducted using six gene loci as proposed previously[51]. An in-house bioinformatic workflow for WGS-based MLST analysis was developed, enabling the determination of DSTs from 629 *C. tropicalis* genomes. For five out of the six loci (*ICL1*, *MDR1*, *SAPT2*, *SAPT4*, and *ZWF1a*), we directly extracted the SNPs within the

reference regions of each locus from the VCF file of the *C. tropicalis* genomes. However, two copies of the *XYR1* gene were found in each haploid chromosome set of *C. tropicalis* by long-reads sequencing. To enhance the comparability of results between WGS-based and Sanger-based MLST analysis, a separate read mapping process was performed, aligning each sample's sequencing reads against a single copy of the *XYR1* reference region sequence to identify SNPs within this locus. Next, the DNA sequences for each locus were reconstructed by a custom script and cross-referenced with data from the PubMLST database (http://www.pubmlst.net, last accessed on Dec 31, 2022) to determine the corresponding gene allele numbers and strains' DSTs. For any novel DSTs identified in this study, DST numbers starting from DST10001 were assigned. It's worth noting that our study revealed discrepancies in the MLST results obtained via the WGS-based workflow when compared to those from prior studies for certain strains. These disparities can be attributed to the bioinformatic pipeline utilized, including differences in mapping approaches (especially concerning the multi-copy *XYR1* locus), variant caller settings, and the algorithm for calling heterozygosity and homozygosity sites.

A ML tree for the comprehensive dataset of 1571 strains was generated, using IQ-TREE (version 1.6.12) with 1000 ultrafast bootstrap replicates, based on concatenated six-locus DNA sequences. A bootstrap criterion of ≥70 was applied to support the determination of MLST clades. For phylogenetic clades that had been previously described in previous publications, the same nomenclature was adopted as provided in these works[13,39]. Newly identified MLST clades were named using the designations of Clade N1 to Clade N8 (Supplementary Data 1). In addition, the MLST allelic profiles of strains were analyzed by goeBURST in PHILOVIZ (version 2.0) software to define clonal complexes (CCs, started from CC0 and named in descending order based on the number of isolates within each CC)[52].

### Phylogenetic and population structure analysis

IQ-TREE was used to infer an ML tree of the 629 isolates using the dataset of 1,642,574 confident SNPs and 1000 ultrafast bootstrap replicates[53], and the script "vcf2phylip.py" (https://github.com/edgardomortiz/vcf2phylip) was used to translate SNPs in VCF format to FASTA format, with heterozygous SNP loci converted as IUPAC bases. The Edge-linked Partition Model was utilized to determine the most suitable DNA model, and branch supports were assessed using 1000 ultrafast bootstrap replicate[54,55]. A bootstrap value of >99.0 was implemented to support the designation of clusters (Fig. 5). Additionally, average nucleotide identity (ANI) was calculated by FastANI (version 1.33) using the genome after quality control (see Genome assembly, quality control and annotation), and the intra-group cutoff value was set as ≥99.6 for proposing WGS clusters (Supplementary Fig. S1)[56]. Whole-genome SNP-based Bayesian Information Criterion (BIC) and ADMIXTURE (version 1.3) analyses were employed to estimate the best number of populations (Supplementary Fig. S2)[57,58]. Furthermore, Principal Components Analysis (PCA) and Discriminant Analysis of Principal Components (DAPC) were carried out by GCTA (version 1.94) and PLINK (version 1.9) to evaluate the inference of genomic clusters (Supplementary Fig. S2)[58-60]. Besides, loss of heterozygosity (LOH) analysis was carried out on 629 genomes as previously described[10].

### Analysis of polyploidy, aneuploidy, large structure variation and copy number variation (CNV)

Polyploidy was detected using the frequency of the non-reference allele for cumulative heterozygous biallelic SNPs across all scaffolds, as described previously[12]. To perform aneuploidy, large structure variation and CNV analysis, a "Splint" algorithm previously established[61] was employed to process the BAM file with 500-bp window frames. This algorithm was designed to mitigate the "smiley pattern" bias commonly observed in the terminal regions of chromosomes[61]. In addition,

the SAMtools "depth" command was used to calculate the sequencing depth of each base from the sorted BAM file of each sample, to predict breakpoint sequences for *ERG11* CNV segments. Further, the mean read depth across nonoverlapping 1-kb windows spanning the entire genome was visualized using a custom script for assisting analysis of aneuploidy and large structure variation events.

### Genome assembly, quality control and annotation

IDBA-hybrid (version 1.1.3) was used to complete de novo assembling of short-read sequencing reads[62]. IDBA's "fq2fa" program was used to convert the filtered FASTQ files to FASTA files, and then *C. tropicalis* MYA-3404 genome was used as a reference with the option "−reference". Canu (version 2.2) was used to assemble the long reads of PacBio[63]. The genome sequences were evaluated for assembly quality using BUSCO (version 5.2.2) with genomic mode[64]. The "−augustus" and "−augustus_species" parameters were added to specify the *C. tropicalis* evaluation model, and saccharomycetes_odb10 served as the database during the evaluation. AUGUSTUS (version 3.3.2) was used for structural annotation with the following options: "−strand=both", "−genemodel = complete" and "−species= candida_tropicalis"[65]. Next, snpEff (version 5.1.2) was utilized for genome annotation, employing the VCF file generated by GATK HaplotypeCaller and the GFF file generated by AUGUSTUS[66], with "codonTable" set to "Alternative_Yeast_Nuclear" as *C. tropicalis* belongs to the CUG clade. BUSCO was again used to evaluate the quality of the predicted CDS sequences using the option "-m transcriptome".

### Pan-genome and core genomic analysis

OrthoFinder (version 4.2.2) was used to screen orthologous genes based on the predicted CDS sequences. The screening process adds the "-d" parameter to specify the input file as a nucleic acid sequence. In the orthologous groups after clustering, the longest and shortest sequences of each homologous group (OGs) were selected as representative sequences. They were used to calibrate results with the genomes of each sample with the Basic Local Alignment Search Tool (BLAST), using an E-value truncation of $1 \times 10^{-5}$ [67]. To reveal unique OGs carried by a certain population, cutoff values of >90% in the studied population and <20% in comparative populations were used. For selected OGs, eggNOG-mapper (version 2)[68] and the NCBI nr database were used to perform functional annotation.

### Antifungal susceptibility testing

For clinical isolates collected in China, antifungal susceptibility testing was carried out using the "gold-standard" broth microdilution method in accordance with CLSI guidelines[69]. Nine commonly used antifungal agents were tested, including four azoles (fluconazole, voriconazole, itraconazole, posaconazole), three echinocandins (caspofungin, micafungin, anidulafungin), 5-flucytosine, and amphotericin B. Susceptibility testing results were interpreted followed the clinical breakpoints (CBPs) or epidemiological cutoff values (ECVs) recommended in the CLSI documents[69,70]. Specifically, for general isolates, the range of fluconazole concentrations tested was 0.12–256 mg/L; however, to explore the potential dose-dependent relationship between amplified copy numbers of the *ERG11* gene and elevated fluconazole susceptibilities, we expanded the upper limit of fluconazole concentration to 1024 mg/L for testing cluster AZR and group AZR-ADJ isolates. Strains *Candida parapsilosis* ATCC 22019 and *Candida krusei* ATCC 6258 were employed as quality controls[69].

### Real-time quantitative PCR

The gene expression levels of the *ERG11* gene (azole drug target), *UPC2* gene (encoding a transcription factor of *ERG11*), as well as *CDR* and *PDR* genes (encoding drug efflux pumps), were examined using real-time quantitative PCR (qPCR) methodology, with the *ACT1* gene used as the internal control. Isolates tested including all cluster AZR isolates

($n = 73$) and all group AZR-ADJ isolates ($n = 26$) collected in China, and a subset of other phylogenetic cluster strains ($n = 15$) as comparators. *C. tropicalis* strain ATCC750 was used as a reference. Briefly, all selected isolates were grown in yeast peptone dextrose medium at 28 °C for 48 h. RNA extraction was performed with RNeasy Mini Kit (Qiagen: 74106). Gene expression levels were evaluated using the comparative CT method[71].

### In vitro induction of *ERG11* gene copy number increase

To validate the dose-dependent correlation between the increase in *ERG11* gene copy number and the escalation of azole MICs, we conducted an in vitro passaging experiment under fluconazole stress. A cluster AZR strain F4082 (*ERG11* copy number = 3, fluconazole MIC = 128 mg/L) was selected. The strain was initially incubated on SDA agar plates (Oxoid), and then a single colony was transferred to an antifungal-free liquid YPD medium (BD Difco: 242720) and incubated at 30 °C for 24 h to establish the 0th passage. An adequate volume of the culture suspension was harvested and subjected to low-speed centrifugation ($1000 \times g$) to remove the supernatant. The cells were then resuspended and inoculated into 3 mL of liquid YPD medium with a fluconazole concentration of 256 mg/L. The inoculum was adjusted to an optical density of 0.05 (corresponding to a cell concentration of $0.9$–$1.4 \times 10^5$ CFU/mL) and then incubated at 30 °C in a rotating shaker at 220 rpm for 24 h. This process was repeated until the 10th passage was achieved. The 0th, 3rd, 6th, and 10th passages were selected to evaluate alterations in antifungal susceptibility (by broth microdilution assay), *ERG11* expression level (by qPCR), and *ERG11* CNVs (by short- and long-read sequencing). Besides, the 10th passage obtained was further passaged for an additional 10 times without exposure to azole stress, in order to assess the stability of *ERG11* CNVs in vitro-induced.

### Statistical analysis

All statistical analyses were performed using R (version 3.6.362). A chi-square or Fisher's exact test was used to examine associations between categorical variables where appropriate, and Bonferroni correction was carried out when necessary. The Kruskal-Wallis H test was carried out on continuous variables where appropriate, and multiple comparisons were carried out if necessary. Spearman's rank correlation was used to evaluate the dependence between the rankings of variables. A *p*-value of <0.05 was considered significant.

### Ethics declarations

This study was approved by the Human Research Ethics Committee of Peking Union Medical College Hospital (no. S-263). Waiver of informed consent was granted as the research only involved the use of fungal isolates and there was no linkage to patient data or identifiable information.

### Reporting summary

Further information on research design is available in the Nature Portfolio Reporting Summary linked to this article.

## Data availability

New short-read and long-read sequencing results of *C. tropicalis* isolates generated in this study have been deposited at the NCBI Sequence Read Archive under BioProject ID PRJNA946688. Additional previously published WGS and MLST data used in this study (Supplementary Data 1 and 2) were sourced from the NCBI SRA database (https://www.ncbi.nlm.nih.gov/sra) and the PubMLST database (http://www.pubmlst.net), respectively.

## Code availability

Custom scripts used in this study have been released at https://chifnet.microonline.cn/NGS_ctr.

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

## Acknowledgements

This study was supported by National Key Research and Development Program of China (grant no. 2022YFC2303002 to M.X.), National High Level Hospital Clinical Research Funding (grant no. 2022-PUMCH-C-052 to M.X.), National Natural Science Foundation of China (grant no. 81802042 to X.F.), and Beijing Hospitals Authority Youth Programme (grant no. QML20190301 to X.F.). We extend our sincere appreciation to all the principal and co-principal investigators from the hospitals that took part in the CHIF-NET study for their significant contributions in collecting the isolates used in this study (Supplementary Data 5). We also extend our gratitude for the support provided by the Clinical Bio-bank, Peking Union Medical College Hospital in sample collection, storage, and management.

## Author contributions

X.F., M.X., J.G. and Y.-C.X. designed the study. X.F., R.-C.D., S.Z., Y.-Y.G., M.K., D.-W.G., Y.-N.M., Y.-H.P. and Z.-Y.S. performed the experiments. X.F., R.-C.D., J.G. and M.X. carried out data analysis and interpretation. X.F., R.-C.D. M.X. J.G. and Y.-C.X. drafted the manuscript. All authors reviewed and approved the manuscript.

## Competing interests

The authors declare no competing interests.
