## [Peer Review File · Nature Communications]

Tandem gene duplications contributed to high-level azole resistance in a rapidly expanding *Candida tropicalis* populationREVIEWER COMMENTS

Reviewer #1 (Remarks to the Author):

Candida tropicalis is becoming more of a public health concern globally, with mortality rates in excess of 50%, and rising levels of azole drug resistance. This paper from Fan et al describes rapid expansion of a drug resistant sublineage, that is also enriched for retrotransposons.

My main comment is that whilst I am convinced about the AZR cluster, I am not convinced about the conclusion of 14 'major' phylogenetic clusters'. It isn't clear from the ADMIXTURE analysis (perhaps inclusion of the BIC curve would help), as Supp Fig 1 is incredibly busy. Figure 2, which is based on MLST, seems to arbitrarily assign isolates to clades, with some isolates appearing intermediate, and not assigned to any cluster. It is not immediately apparent from the WGS phylogeny (Figure 5) that there are 14 clusters either. Standard population genomic analysis would first look at bootstrap support for clusters (indeed, there is no phylogeny in this manuscript indicating what the bootstrap support is - this should be included somewhere, even if it's just a supplementary figure), then multivariate analyses (PCA, DAPC, STRUCTURE, ADMIXTURE) to confirm. As it stands, I do not see compelling evidence for the conclusion of 14 clusters.

Whilst I appreciate my next comment can only be addressed by substantial lab work, it must be said that this paper addresses potential mechanisms, and does not confirm the contribution of duplications, retrotransposons, or CNVs to high-level azole resistance. Whilst it is acknowledged in the discussion that further work is needed to test these associations, I feel it is too bold to claim that high-level azole resistance is caused due to these mechanisms without confirmation (see lines 452-453 in the discussion that states AZR resistance is due to tandem gene duplications of ERG11 containing A395T).

Minor comments:

References/citation of A395T mutation conferring resistance needed throughout 'custom scripts' in the methods (lines 513 and 539) need to be made available in accordance with Nature publishing guidelines

Annotation used for snpEff needs to be states

Figure 2 needs a scale

Figure 3: words slightly cut off in a and b

Figure 5: can you add a black outline around the square for the breakpoint MIC for resistance for fluconazole and voriconazole - this would help non-specialist readers understand which MICs refer to susceptible and resistant without looking it up.

Reviewer #2 (Remarks to the Author):

The NCOMMS-23-13862-T manuscript entitled "Tandem gene duplications led to high-level azole resistance in a rapidly expanding *Candida tropicalis* population" describes that cluster AZR isolates exhibited a distinct high-level azole-resistance due to tandem duplications of the ERG11A395T mutant gene allele. This is an interesting observational finding and it is suitable for other journals.

To publish on the journal of Nature Communication, the authors need to provide more information as following:

1. In order to prevent the selection/spread of AZR, the authors can investigate:
 - a. whether AZR can be converted back to susceptible?
 - b. Whether the tandem duplication of ERG11 a key contributor for azole resistance of AZR? For example, will the cells become azole-susceptible when the copy of ERG11 of AZR back to single copy?
2. Even though the authors provide some genes may be involved in ERG11 CNV, it would be necessary to identify which gene(s) involved in regulating ERG11 CNV. To understand the mechanism may help to design strategy to address the treats posed by AZR.

3. The minor concern is that if ERG11 of all tested clade 4 has two mutations, Y132F and S154F, why the authors emphasized Y132F only.

Reviewer #3 (Remarks to the Author):

The authors contribute a large repository of *C. tropicalis* isolates to the WGS data collection with MLST findings and susceptibility data. The work will be of great significance to the field and contributes to the understanding of CNV involving ERG11 and the mechanism.

Some of the conclusions are over-reaching, and the discussion should reflect and comment on sampling bias. Note should be made of any known epidemiological links and whether a known outbreak constituted the basis for collecting fluconazole resistant isolates. Suggestion has been made in the literature of a link to agriculture - this should be commented on with any linked DST by MLST.

Substantial revision is required to clarify and justify the methodology. There is insufficient detail for the work to be reproduced.

INTRO

Line 82 - Correct "Austria and Singapore" to "Australia and Singapore"

METHODS

How were the 450 isolates selected? There was evidently a bias towards fluconazole resistance. A brief description of criteria for selection (sequential? Sample of convenience including all resistant strains and some susceptible?) should be included in methods.

What was the susceptibility testing method and interpretation criteria?

QPCR was done, but is not described in the methods. This must be included and correlated with the section on CNV.

Line 461 - Only data from one study with international isolates was included. What were the reasons for not using all *C. tropicalis* isolate WGS data deposited in NCBI?

How were the data from (12) included? Could additional susceptibility data obtained from those authors? Were these 78 isolates included in some but not all of the analysis?

It appears that susceptibility data from these isolates has been left blank in Table S2. This should be stated in the methods.

Note that other WGS data available in NCBI exist that are linked with MIC data.

Line 473 - how many of the isolates were from bloodstream?

Line 484 What read depth was aimed for and what was accepted for short read sequencing? As this is a diploid species, greater read depth is necessary eg x60-x100. A read depth of x20 means only x10 for each allele and reduces the confidence of calls. A supplementary file with read depth for each isolate and other sequencing metrics would be helpful.

Line 484-488 Long read sequencing was performed. Was Pacbio sequencing performed on all isolates? The results section suggests it was only done on 5 isolates - this should be stated in the methods

Line 495-499 What was the threshold for calling a heterozygous SNP? Eg 20% of the alternative call in reads? 30%?

Line 516-517 how were heterozygous sites handled in phylogenetic analysis - were these masked? Were the two versions concatenated?

Line 528-531 Were these MLST calls submitted to PubMLST for curators to check? was there anchoring of MLST call with a previously reported isolate? How were new diploid sequence types assigned? Were they given an in-house DST, or assigned by PubMLST?

Line 533-535 How were the clonal complexes (CC) numbered from goeBURST in PHILOVIZ - did they start at 0 or at 1?

Line 533 - "MLST clades were assigned with reference to previous publications.13, 35" How were clades defined? What program was used? If it was an in-house script, what were the criteria?

The previous publications do not describe how a Clade was assigned. I do note that they both included the curator of the *C. tropicalis* MLST database. The description given in previous publications is "clades containing more than 10 genetically closely related DSTs were labelled." The paper referred to by (13) is one that uses eBurst analysis for clonal clusters so it is unclear how clades were derived.

13 -Tseng KY, Liao YC, Chen FC, Chen FJ, Lo HJ. A predominant genotype of azole-resistant *Candida tropicalis* clinical strains. *Lancet Microbe* 3, e646 (2022).

35 - Zhou ZL, et al. Genetic relatedness among azole-resistant *Candida tropicalis* clinical strains in Taiwan from 2014 to 2018. *Int J Antimicrob Agents* 59, 106592 (2022).

Which MLST alleles were found to be multicopy? Long read sequencing would have made this easy to identify. How were these accounted for /handled? Was one version masked?

Line 538-539 How was the bias of library prep accounted for? Nextera DNA prep was used. The amplification tends to bias read depth at the ends which can artificially create the impression of CNV.

Line 543-544 How does this method compare to the method Ymap? Biases have not been discussed in this section. I recommend anchoring some findings of CNV by running them through the Ymap pipeline. Note the this relevant section in their publication

<https://genomemedicine.biomedcentral.com/articles/10.1186/s13073-014-0100-8>

"Several biases can impact read depth and thereby interfere with CNV analysis. Two separate biases, a chromosome-end bias and a GC-content bias, appear sporadically in all types of data examined (including microarray and whole genome sequencing (WGseq) data). The mechanism that results in the chromosome end artifact is unclear, but the smooth change in the apparent copy number increase towards the chromosome ends (Figure 2A) suggests that some DNA preparations may release more genomic DNA as a function of telomere proximity (Jane Usher, personal communication). A GC-content bias is due to strong positional variations in GC content in the *C. albicans* genome. This, combined with the PCR amplification bias introduced during sequence library or array preparation, results in a strong positional effect in local copy number estimates"

Was the method used in this manuscript also able to demonstrate the increased copy number demonstrated in the isolates by McTaggart using Ymap ?

RESULTS

How many new DSTs were described in the new study isolates? A description of MLST findings with a breakdown by allele is missing.

Line 138-140 This needs to be rephrased. The data are heavily geographically biased and not representative 'worldwide'

Lines 141-142 Include denominator with %, so that it is clearer that the percentage refers to all the data included, >80% of which is from China

Line 144 - This is a very high rate of resistance which should be put into context of isolate selection "The overall rates of fluconazole resistance were 32.1% (n = 472) in the 1,470 isolates"

Line 149 - suggest removing the word 'led' and the word 'worldwide'. There is selection bias.

Lines 154-159 include location

Line 180 - "aneuploidies were found on chromosome 6 (n = 5), chromosome 5 (n = 3)" - this is unexpected as aneuploidies are more commonly found on chromosome 5. What was done to check this?

Line 183 - "genome-wide high levels 183 of heterozygosity were observed in 3.2% (17/528)" were these assessed against fungal database eg ITS database for hybrid species?

Line 184-185 These needs to be rephrased and qualified "However, no associations were observed between polyploidy, aneuploidy, large structural variation or high heterozygosity genotypes and antifungal resistance."

One would only expect these to be associated with antifungal resistance if involving Chromosome 5 as alluded to in the introduction. The only isolate in Table S2 with aneuploidy involving chromosome 5 and susceptibility data is fluconazole resistant. The other isolates with aneuploidy in chromosome 5 have susceptibility data missing. It is also not clear from Figure 4 or Table S2 which isolates were defined as having a different ploidy, and what the fluconazole susceptibility

was. This data needs to be included in S2, also consider including in this paragraph.

Line 191 - correct 'shown' as ungrammatical in this sentence "WGS phylogenetic clusters shown good correlations with MLST clades"

Line 193 - remove 'of the' in this phrase "except for those isolates not assigned to any of the MLST clade"

Line 194 - this requires explanation "CTC10, which was made up of isolates from MLST clades 3 and 6 that were phylogenetically divergent"

Line 212 - either add 'the' or change 'none of' to 'No' in this phrase "None of fluconazole-susceptible isolates"

Line 217-218 What does this mean? "There was also a significantly higher proportion of AZR-ADJ isolates that were ERG11 gene-homozygous (50.0%)" Homozygosity/heterozygosity always varies across the ERG11 gene. Is this implying that the ERG11 gene was heterozygously present/absent? This would be unusual.

Line 243 - This is not surprising as it is consistent with what the literature has suggested for *C. albicans* and *tropicalis* "Surprisingly, functional annotation indicated that this ORF was the ERG11 gene."

Line 250 - "breakpoint sequences were the same in all cluster AZR isolates." Was there any evidence of an epidemiological link between isolates? Was this a set of outbreak strains that would be expected to be clonally related? What was the SNP difference between these isolates?

Line 264-265 This is very interesting. Are these mutations necessary for duplication events? If so what explains the increased copy number in the ERG11 gene in the McTaggart isolates? "tandem gene duplication events had only occurred on the chromosome with mutated ERG11 alleles with the A395T and C461T mutations,"

Line 285 This must be described in the methods "of the ERG11 gene, qPCR was carried out within a subset of cluster AZR isolates (n=29)"

Reviewer Comments & Author Responses

Reviewer #1

Candida tropicalis is becoming more of a public health concern globally, with mortality rates in excess of 50%, and rising levels of azole drug resistance. This paper from Fan et al describes rapid expansion of a drug resistant sublineage, that is also enriched for retrotransposons.

Response: We would like to thank the reviewer for going through our manuscript and for the thorough comments. We are delighted that the reviewer highlighted the significance of azole-resistance in *C. tropicalis* and emphasized the key findings of our study.

1. My main comment is that whilst I am convinced about the AZR cluster, I am not convinced about the conclusion of 14 'major' phylogenetic clusters'. It isn't clear from the ADMIXTURE analysis (perhaps inclusion of the BIC curve would help), as Supp Fig 1 is incredibly busy. Figure 2, which is based on MLST, seems to arbitrarily assign isolates to clades, with some isolates appearing intermediate, and not assigned to any cluster. It is not immediately apparent from the WGS phylogeny (Figure 5) that there are 14 clusters either. Standard population genomic analysis would first look at bootstrap support for clusters (indeed, there is no phylogeny in this manuscript indicating what the bootstrap support is - this should be included somewhere, even if it's just a supplementary figure), then multivariate analyses (PCA, DAPC, STRUCTURE, ADMIXTURE) to confirm. As it stands, I do not see compelling evidence for the conclusion of 14 clusters.

Response: We appreciate the reviewer's suggestion, and WGS-based population genomic analysis were carried out again, following the standard procedure as recommended by the reviewer.

Firstly, we incorporated a bootstrap value threshold of >99.0 to provide strong support for identifying WGS clusters, and we have clearly indicated the related bootstrap values in our revised Fig. 5. Additionally, we conducted an average nucleotide identity (ANI) analysis with a stringent threshold of >99.6% (see new Supplementary Fig. 1) for proposing phylogenetic clusters. To determine the optimal number of clusters, we also performed Bayesian Information Criterion (BIC) and ADMIXTURE best K analyses, both of which indicated the presence of $n = 36$ populations (presented in the new Supplementary Fig. 2). PCA and DAPC analyses also supported these findings (Supplementary Fig. 2).

While out of the 36 identified clusters, we have decided to name only the 22 major clusters that have a minimum of five isolates as CTCs (including cluster AZR and group AZR-ADJ) in this study. We have re-performed all cluster-related analyses throughout our manuscript, and revisions were made to all relevant contents in Results and Discussion sections (e.g., Lines 195-199, revised Fig. 4). Furthermore, we have updated the Methods section to reflect related improvements of our methodology (Lines 649-663).

2. Whilst I appreciate my next comment can only be addressed by substantial lab work, it must be said that this paper addresses potential mechanisms, and does not confirm the contribution of duplications, retrotransposons, or CNVs to high-level azole resistance. Whilst it is acknowledged in the discussion that further work is needed to test these associations, I feel it is too bold to claim that high-level azole resistance is caused due to these mechanisms without confirmation (see lines 452-453 in the discussion that states AZR resistance is due to tandem gene duplications of *ERG11* containing A395T).

Response: We appreciate the suggestion provided by the reviewer. In order to gain deeper insights into the underlying mechanism of *ERG11* CNVs and its association with high-level azole resistance, we conducted additional laboratory works for exploration. The first one aimed to investigate a potential dose-response relationship between *ERG11* CNV, *ERG11* gene expression level, and azole susceptibility. Of note, the previous upper limit of fluconazole concentration (256 mg/L) was insufficient for this analysis. Therefore, we expanded the upper limit to 1024 mg/L and re-evaluated the fluconazole susceptibility of all cluster AZR isolates. Additionally, we tested the *ERG11* gene expression levels of all cluster AZR and group AZR-ADJ isolates. Our findings revealed a notable correlation between increased *ERG11* copy number and elevated gene expression level, which accompanied by a higher distribution of azole minimum inhibitory concentrations (MICs). We have incorporated this new information into the relevant sections of the manuscript (Lines 295-305, 314-323 and revised Fig. 7).

Furthermore, we conducted an additional *in vitro* experiment, inducing a cluster AZR strain (initial *ERG11* copy number = 3, fluconazole MIC = 128 mg/L) under fluconazole stress conditions (concentration of 256 mg/L). After 10 passages, we observed a significant increase in the *ERG11* copy number (to n = 7), along with a 2-fold elevation in both *ERG11* gene expression level and fluconazole MIC. These findings provide additional compelling evidence supporting the association between the *ERG11* CNV mechanism and high-level fluconazole resistance in cluster AZR isolates. The detailed description of these results and related methods have also been involved in the revised manuscript (Lines 340-359 and 724-738, new Fig. 9).

However, we acknowledge that there is still knowledge for further investigations. For instance, we have yet to establish a direct link between enrichment of retrotransposons and presence of *ERG11* CNVs, as well as azole resistance. Consequently, when drawing our conclusions, we have taken a more cautious approach and revised related contents accordingly, as indicated by the reviewer (Lines 556-557).

3. Minor comments: References/citation of A395T mutation conferring resistance needed throughout

Response: We have included relevant citations in our manuscript (e.g. at Lines 88-91 and 224-226)

4. 'custom scripts' in the methods (lines 513 and 539) need to be made available in accordance with Nature publishing guidelines

Response: We have made the custom scripts generated during this study publicly available, in accordance with Nature publishing guidelines (declaration has been made at Lines 750-751).

5. Annotation used for snpEff needs to be states

Response: Details of the snpEff annotation have now been included in Methods section (Lines 685-688).

6. Figure 2 needs a scale

Response: For Fig. 2, an unrooted layout with equal-angle method was used to visualize the MLST maximum-likelihood tree. As branch lengths are not proportional to evolutionary distances or genetic differences between populations, the layout does not have a branch length scale. We have included the description in the revised legend for Fig. 2 (Line 1010).

7. Figure 3: words slightly cut off in a and b

Response: We have revised Fig. 3 to address the issue (new Fig. 3).

8. Figure 5: can you add a black outline around the square for the breakpoint MIC for resistance for fluconazole and voriconazole - this would help non-specialist readers understand which MICs refer to susceptible and resistant without looking it up.

Response: Thanks for the valuable suggestion from the reviewer. We have implemented some modifications to better illustrate the categorical interpretations of fluconazole and voriconazole susceptibility for our readers (new Fig. 5).

Reviewer #2

1. The NCOMMS-23-13862-T manuscript entitled "Tandem gene duplications led to high-level azole resistance in a rapidly expanding *Candida tropicalis* population" describes that cluster AZR isolates exhibited a distinct high-level azole-resistance due to tandem duplications of the *ERG11*^{A395T} mutant gene allele. This is an interesting observational finding and it is suitable for other journals.

Response: We would like to thank the reviewer for evaluating our study. We have conducted additional experiments and revised the manuscript based on the suggestions from the reviewer. However, we would like to express our viewpoint that we believe the significance of this research aligns with the scope of Nature Communications and is relevant to the interests of the readers.

Nowadays, there is a concerning rise in the prevalence of deadly fungal infections, and among them, *C. tropicalis* stands out as a notable fungal pathogen with superior mortality rates and increased resistance to antifungal treatments. Unfortunately, the challenges posed by fungal infections are often overlooked due to our insufficient understanding of fungal pathogens.

Through an extensive genetic population study comprising global *C. tropicalis* strains, our research has made two significant contributions. Firstly, we observed the emergence of a highly azole-resistant *C.*

tropicalis population, nomenclated as cluster AZR in our study, which has widely disseminated in the Asia-Pacific region. Moreover, we found population expansion of this cluster was directly associated with a substantial increase in azole resistance rates in China, rising from less than 6% to over 30% within a decade. Secondly, our study has revealed a distinct genetic feature in cluster AZR isolates - the tandem gene duplication of the *ERG11* gene. Further laboratory investigations have confirmed that this mechanism was strongly correlated with the high-level azole resistance exhibited by the population. Our findings indicate that the AZR population demonstrates enhanced adaptability in response to selective pressure from antifungal agents, which is believed to be associated with the rapid population expansion observed, leading to a substantial increase in the rate of azole resistance.

By uncovering the presence of cluster AZR and elucidating the genetic basis of its resistance, our study not only highlights the urgent need for addressing fungal infections but also offers valuable insights into the development of strategies to curb the spread of azole-resistant *C. tropicalis*. The implication of our study extends beyond the boundaries of medical and microbiological research. Considering the global reach and influence of Nature Communications, we firmly believe that our study aligns perfectly with the interests and concerns of its diverse readership.

2. To publish on the journal of Nature Communications, the authors need to provide more information as following: In order to prevent the selection/spread of AZR, the authors can investigate:

a. Whether AZR can be converted back to susceptible?

b. Whether the tandem duplication of *ERG11* a key contributor for azole resistance of AZR? For example, will the cells become azole-susceptible when the copy of *ERG11* of AZR back to single copy?

Response: We express our gratitude to the reviewer for the valuable feedback, which prompted us to conduct further laboratory research.

To address the question of whether the tandem duplication of the *ERG11* gene is a key contributor for azole resistance of AZR, we performed two additional experiments, specifically focusing on determining the correlation between *ERG11* CNVs, *ERG11* gene expression levels, and azole susceptibilities. Firstly, we conducted an additional analysis within all cluster AZR isolates collected in this study, and revealed that an increased *ERG11* copy number was associated with elevated *ERG11* gene expression level and enhanced azole resistance. Secondly, we carried out an *in vitro* experiment to induce a cluster AZR strain that initially exhibited a low *ERG11* copy number (n = 3) under fluconazole stress. We observed a significant increase in *ERG11* CNV (reaching up to n = 7) in the 10th passage of the strain, meanwhile there was a substantial rise in both *ERG11* gene expression level and fluconazole MIC. These experiments provided further validation of a dose-response relationship between *ERG11* CNV, *ERG11* gene expression level, and azole susceptibility, supporting that *ERG11* plays an important role to reduced azole susceptibility, specifically high-level azole resistance, in cluster AZR isolates. These changes were

also described in our responses to Reviewer 1's Comment 2. Additional results were described e.g. at Lines 295-305, 314-323, 340-351 and illustrated in a new Fig. 9.

For whether AZR can be converted back to susceptible, we also tried to passage the original strain used in above-described *in vitro* experiment (0th passage) and its 10th passage for 10 times without subjecting to fluconazole exposure. Our results showed no changes in both the fluconazole MICs and the copy numbers of *ERG11* gene. This indicated once *ERG11* CNVs has emerged, it generally maintained stable and persisted in the absence of external azole stresses. We have integrated these results into our manuscript (Lines 353-359).

Regarding the query raised by the reviewer regarding if “the cells become azole-susceptible when the copy of *ERG11* of AZR back to single copy”, we think that observing such a genotype alternation under natural conditions, or by *in vitro* passaging would be challenging. However, *C. tropicalis* is not a well-studied model organism, and available genetic research tools for *C. tropicalis* are not yet well-developed. Therefore, our research team is presently unable to reverse cluster AZR strains with *ERG11* tandem gene duplications back to a single-copy genotype by laboratory assays. We acknowledge this as a limitation in our work, which was discussed in Lines 532-536.

Nevertheless, we believe that our laboratory findings, which demonstrate a dose-dependent correlation between *ERG11* CNVs and azole MICs, already offer compelling evidence of the significant role played by *ERG11* CNV in driving high-level azole resistance in the clustered AZR strains.

3. Even though the authors provide some genes may be involved in *ERG11* CNV, it would be necessary to identify which gene(s) involved in regulating *ERG11* CNV. To understand the mechanism may help to design strategy to address the treats posed by AZR.

Response: We appreciate the reviewer for the comment. As is widely recognized, CNV is a significant source of genetic diversity in driving evolutionary processes, which have the potential to contribute to important phenotypic variations. To date, several different replication and repair mechanisms have been shown to be involved in the development of CNV, such as non-allelic homologous recombination (NAHR), microhomology-induced replications, and unequal crossovers. However, the understanding of the regulatory mechanism of CNV, including key genes involved in this process, is still limited, even in human and other well-studied microorganisms.

Although efforts have been made in this study, we did not identify a direct regulatory mechanism associated with *ERG11* CNV. Instead, our pangenome research revealed the enrichment of retrotransposons, specifically the Ty3/gypsy-like retrotransposons in cluster AZR isolates. This finding suggests that the genomes of cluster AZR strains may exhibit higher plasticity. Unfortunately, we were unable to confirm a direct mechanism correlation between enrichment of retrotransposon and *ERG11* CNVs. We have included this aspect as a limitation of our study in Lines 536-537.

We agree with the reviewer's concerns regarding the urgent need to design strategies aimed at preventing the selection and spread of azole resistance. As one of the most significant contributions of our study, we for the first time illustrated that the growing antifungal resistance threats in the Asia-Pacific regions were predominantly attributed to the rapid expansion of cluster AZR. Based on the findings, we could propose the implementation of targeted monitoring for the cluster AZR population, as it can serve as a valuable indicator to assess the severity of antifungal resistance challenges over time and evaluate the effectiveness of transmission control measures. While specific measures targeting the transmission of a particular clone can be valuable, we believe that a more practical approach to curbing resistance lies in reducing azole stresses in clinical settings and the environment, particularly our *in vitro* experiment has demonstrated that cluster AZR can acquire a stronger resistant phenotype under azole stress through the *ERG11* CNV mechanism.

Therefore, though we may not have explained the mechanisms regulating *ERG11* CNVs, we believe this does not diminish the significant implications of our study's findings. The above contents have been incorporated into Discussion section of our manuscript at Lines 537-545.

3. The minor concern is that if *ERG11* of all tested clade 4 has two mutations, Y132F and S154F, why the authors emphasized Y132F only.

Response: Thanks for the reviewer's question.

It has been verified that the *ERG11*p substitution Y132F (led by gene mutation A395T/W) play a crucial role in conferring azole resistance in *C. tropicalis*. Meanwhile, although substitution S154F (resulted from mutation C461T/Y) is often observed alongside the Y132F substitution, previous studies have shown that this mutation does not independently contribute to the azole-resistant phenotype. Therefore, our description primarily focused on the Y132F substitution in our manuscript. We have incorporated relevant explanations into our manuscript (Lines 226-228).

Reviewer #3

The authors contribute a large repository of *C. tropicalis* isolates to the WGS data collection with MLST findings and susceptibility data. The work will be of great significance to the field and contributes to the understanding of CNV involving *ERG11* and the mechanism.

The authors contribute a large repository of *C. tropicalis* isolates to the WGS data collection with MLST findings and susceptibility data. The work will be of great significance to the field and contributes to the understanding of CNV involving *ERG11* and the mechanism.

Some of the conclusions are over-reaching, and the discussion should reflect and comment on sampling bias. Note should be made of any known epidemiological links and whether a known outbreak constituted the basis for collecting fluconazole resistant isolates. Suggestion has been made in the literature of a link to agriculture - this should be commented on with any linked DST by MLST.

Substantial revision is required to clarify and justify the methodology. There is insufficient detail for the work to be reproduced.

Response: We thank the reviewer for the positive feedback, and for recognizing the extensive effort we made in our WGS work as well as the significance of our study.

Based on the reviewer's comments, we have made important modifications to enhance the robustness and clarity of our manuscript.

1) To address the reviewer's concern on sampling, we have taken measures of including a more diverse set of global strains in our WGS and MLST analysis to broaden the scope of our findings. Additionally, carefully revisions were made for related results and discussion to avoid overreaching interpretations.

2) In response to the reviewer's comment regarding the connection between azole resistance and agriculture as reported in previous literature, we have incorporated additional discussion and included the reference (Lines 413-416) to provide clarity, that supporting evidences were from molecular typing and phylogenetic analysis methods.

3) We also improved the methodology section to provide more comprehensive and transparent information, aiming to enhance the reproducibility and reliability of our research.

Please find below our point-by-point responses to the reviewer's specific comments and suggestions, which provide more detailed insights into the changes made throughout the manuscript.

1. INTRO: Line 82 - Correct "Austria and Singapore" to "Australia and Singapore"

Response: Thanks, and we have now corrected the misspelling of "Australia" (Line 81).

2. METHODS: How were the 450 isolates selected? There was evidently a bias towards fluconazole resistance. A brief description of criteria for selection (sequential? Sample of convenience including all resistant strains and some susceptible?) should be included in methods.

Response: As recommended by the reviewer, we have incorporated a detailed description of the selection criteria for the 450 isolates in our manuscript (Lines 568-572). As the current study is aiming to gain a comprehensive understanding of potential longitudinal changes in the azole-resistant population, we ensured a consistent proportion of azole-resistant strains (25%-35%) within each surveillance year. However, as pointed out by the reviewer, this approach may lead to the over-selection of fluconazole-resistant strains in earlier years. We have taken note of this limitation and included relevant discussion in our manuscript to avoid potential over-interpretation (Lines 526-531).

3. What was the susceptibility testing method and interpretation criteria?

Response: We have incorporated a new section titled "Antifungal susceptibility testing" within the Methods to describe the methodology we employed. (Lines 701-712)

4. QPCR was done, but is not described in the methods. This must be included and correlated with the section on CNV.

Response: We have now included the description of the qPCR methodology in our manuscript (Lines 714-722).

5. Line 461 - Only data from one study with international isolates was included. What were the reasons for not using all *C. tropicalis* isolate WGS data deposited in NCBI? How were the data from (12) included? Could additional susceptibility data obtained from those authors? Were these 78 isolates included in some but not all of the analysis? It appears that susceptibility data from these isolates has been left blank in Table S2. This should be stated in the methods. Note that other WGS data available in NCBI exist that are linked with MIC data.

Response: We sincerely appreciate the suggestions given by the reviewer. Previously, we only included 78 strains of *C. tropicalis* from O'Brien et al.'s study (old Reference no. 12), because their isolates' collection has covered major continents globally. However, we also recognized that including more strains would enhance the representativeness and generalizability of our study's findings. Following the reviewer's suggestion, we have acquired all publicly available *C. tropicalis* genomes from the NCBI database, and incorporated those formally published in previous literatures into our research.

In total, we included an additional set of 103 strains from eight different publications for WGS analysis (some genomes were excluded due to not meeting the quality control criteria). It brought the total number of strains subjected to WGS analysis to 629 in our work. Simultaneously, we thoroughly updated all antifungal susceptibility results from the literature, including those from O'Brien et al.'s study, and related information was integrated into the updated Supplementary Table S2. Specifically, we would like to clarify that all strains obtained from the NCBI database underwent the complete WGS analysis process, along with the Chinese strains collected (e.g. wgMLST, phylogenomic, pangenome analyses, etc.).

By involving these new strains, the representativeness of our research was enhanced; for instance, the proportion of strains from Oceania in the WGS set increased from 0.1% to 11.5%. Through the comprehensive reanalysis, we also made several additional important discoveries. For instance, we newly identified azole-resistant cluster AZR strains from Australia and Singapore, and these strains also have *ERG11*^{A395T} CNVs. These findings further augment the significance and generalizability of our study.

Due to new strains were involved and all WGS analyses were re-conducted, we have thoroughly revised all related contents from our manuscript (e.g. at Lines 41-43 in Abstract, Lines 130-132 in Introduction, Lines 564-567 in Methods, and all relevant contents in Results and Discussion sections).

6. Line 473 - how many of the isolates were from bloodstream?

Response: There were n = 1,376 clinical isolates with detailed information on isolation sources, 58.1% (n = 782) were obtained from bloodstream infections. We have now added the information at Lines 584-585.

7. Line 484 What read depth was aimed for and what was accepted for short read sequencing? As this is a diploid species, greater read depth is necessary e.g. x60-x100. A read depth of x20 means only x10 for each allele and reduces the confidence of calls. A supplementary file with read depth for each isolate and other sequencing metrics would be helpful.

Response: Thanks for the reviewer's suggestion. During our reanalysis of the WGS data, an average read depth of 60× was considered to provide a high level of confidence for subsequent analyses. Overall, 100% of 458 clinical isolates collected in China, along with 95.6% (173/181) of global strains from previous studies fulfilled this criterion. We have integrated relevant descriptions in the Results and Methods sections (Lines 179-180 and 622-623), and the read depth for each isolate has been incorporated into the revised Supplementary Table S2.

8. Line 484-488 Long read sequencing was performed. Was Pacbio sequencing performed on all isolates? The results section suggests it was only done on 5 isolates - this should be stated in the methods.

Response: Long-read sequencing was conducted to characterize the spatial distribution of *ERG11* CNV segments in genomes. For this purpose, five selected cluster AZR isolates and one AZR-ADJ isolate were tested. We have added this statement to the Methods section at Lines 598-599.

9. Line 495-499 What was the threshold for calling a heterozygous SNP? E.g. 20% of the alternative call in reads? 30%?

Response: Thanks for the reviewer's comments. We would like to clarify that GATK HaplotypeCaller used in our analysis does not rely on fixed thresholds for identifying heterozygous and homozygous SNPs. Instead, the software's model leverages probabilistic methods to calculate genotype likelihoods, which represent the likelihood of each possible genotype at a given genomic position. These likelihoods are then used to assign genotypes, including heterozygous and homozygous variants, based on the most likely genotype for each sample. By utilizing a dynamic and data-driven approach, GATK HaplotypeCaller adapts to the specific characteristics of the sequencing data, leading to improved sensitivity and specificity in variant calling. We hope this clarification assures the reviewers of our SNP calling methodology.

10. Line 516-517 how were heterozygous sites handled in phylogenetic analysis - were these masked? Were the two versions concatenated?

Response: In our study, a publicly available script "vcf2phylip.py" (<https://github.com/edgardomortiz/vcf2phylip>) was used to translate SNPs in VCF format to FASTA format, with heterozygous SNP loci converted using IUPAC nucleotide ambiguity codes. The IQ-TREE software employed for phylogenetic analysis is able to recognize and process these IUPAC codes in the FASTA inputs as heterozygous sites, ensuring that the genetic diversity represented by heterozygous positions is accurately taken into account. We have provided clarification on this methodology in the Methods section of our manuscript (Lines 640-642)

11. Line 528-531 Were these MLST calls submitted to PubMLST for curators to check? was there anchoring of MLST call with a previously reported isolate? How were new diploid sequence types assigned? Were they given an in-house DST, or assigned by PubMLST?

Response: We appreciate the reviewer’s comment. In this study, we developed an in-house workflow for conducting WGS-based MLST on *C. tropicalis* isolates, utilizing the six loci recommended by the PubMLST database. We noticed that our initial manuscript lacked sufficient details on this approach, and our revised version has now provided more detailed descriptions of the methodology (Lines 626-637), and our custom scripts were also released per Nature Publishing Guideline as suggested by Reviewer 1’s Comment 4 (declared at Lines 750-751)

In our analysis, we discovered a total of 201 new DSTs among the genomes of 629 strains, including those (a) had novel allele variants at any of the six loci, or (b) had new combinations of six-locus allelic profiles. Due to the submission requirements of the PubMLST database, which only accepts novel DSTs that have been confirmed through sanger sequencing, we refrained from submitting our novel MLST calls to PubMLST database for validation. In order to conduct further analysis, in-house DST numbers were assigned to these novel DSTs (DST10001 to DST10204, detailed in Supplementary Tables S1 and S2).

Ten isolates included in our WGS-based MLST analysis had previously characterized by traditional Sanger sequencing-based MLST methodology in one of our group's earlier studies, and data from these isolates has been submitted to the PubMLST database. These ten strains were used for the methodology validation analysis. Surprisingly, only 3/10 strains showed identical DST results when analyzed using both methods. For the seven remaining isolates, six exhibited discrepancies in the *XYRI* locus. Moreover, three strains exhibited different results at the *MDRI* locus, and one strain showed a discrepancy at the *ZWF1a* locus. The summary tables below highlight these differences for a clearer understanding.

Table. Ten isolates used for WGS-based and Sanger sequencing-based MLST methodology comparative analysis.

Sample	Method	DST	ICL1	MDRI	SAPT2	SAPT4	XYRI	ZWF1a
F9505	Sanger	321	1	4	12	23	43	9
	WGS	184	1	4	12	23	36	9
F9485	Sanger	322	1	7	12	50	9	3
	WGS	10076	1	7	12	50	54	3
F9486	Sanger	322	1	7	12	50	9	3
	WGS	10076	1	7	12	50	54	3
F9489	Sanger	329	9	90	3	50	24	3
	WGS	329	9	90	3	50	24	3
F9502	Sanger	330	1	44	12	7	48	22
	WGS	10011	1	7	12	7	94	22
F9503	Sanger	331	1	22	12	17	60	22
	WGS	331	1	22	12	17	60	22
F9504	Sanger	332	1	1	1	10	9	3
	WGS	332	1	1	1	10	9	3
F9506	Sanger	333	1	44	12	7	94	22

	WGS	10011	1	7	12	7	94	22
F9507	Sanger	334	9	7	3	3	1	9
	WGS	508	9	7	3	3	54	1
F9508	Sanger	337	1	44	1	7	38	3
	WGS	1361	1	7	1	7	58	3

* Red-labelled cells (with bold-underline font) indicated discrepant DSTs/loci alleles.

Table. A summary of locus with discrepancy results.

Sample	Locus with discrepancy results	Method	Mutations predicted (vs. sequences derived from reference strain MYA-3404)
F9502	MDR1	Sanger	C250T,G262A,G310A, T340K ,A367T,C421T
		WGS	C250T,G262A,G310A,A367T,C421T
F9506	MDR1	Sanger	C250T,G262A,G310A, T340K ,A367T,C421T
		WGS	C250T,G262A,G310A,A367T,C421T
F9508	MDR1	Sanger	C250T,G262A,G310A, T340K ,A367T,C421T
		WGS	C250T,G262A,G310A,A367T,C421T
F9505	XYRI	Sanger	T11Y,C14Y,C242Y,T344Y
		WGS	T11Y,C14Y, T134K ,C242Y,T344Y
F9485	XYRI	Sanger	T11Y,C14Y,C242Y
		WGS	T11Y,C14Y, C188M ,C242Y
F9486	XYRI	Sanger	T11Y,C14Y,C242Y
		WGS	T11Y,C14Y, C188M ,C242Y
F9502	XYRI	Sanger	T59Y,T83Y,A101W,C215Y,C242Y,T344Y
		WGS	T59Y,T83Y,A101W, T134K ,C215Y,C242Y,T344Y
F9507	XYRI	Sanger	T304K
		WGS	No mutation
F9508	XYRI	Sanger	T11Y,C14Y,T59Y,A101W,C242Y,T344Y
		WGS	T11Y,C14Y,T59Y,A101W,C242Y, T287W ,T344Y
F9507	ZWF1a	Sanger	T11Y,C14Y, T59Y , T83Y ,C242Y
		WGS	T11Y,C14Y, C188M ,C242Y

* **Bold-underline font indicated discrepant mutant site.**

Through further comparative analysis, we discovered that the results obtained from both methods differed only by 1-3 bps at each locus, and all discrepancies were observed at heterozygous sites. To ensure accuracy, we meticulously reviewed every Sanger sequencing result manually. Upon thorough examination, we confidently concluded that 100% of these inconsistent results were indeed correct based on the WGS-based method.

The discrepancies observed are attributed to the diploid nature of the *C. tropicalis* species, and Sanger sequencing results require manual checking for heterozygosity at each base, which can lead to the unintentional oversight of certain heterozygous sites. This limitation of MLST has been highlighted in previous studies. The comparative analysis indicated our WGS-based MLST analysis workflow is reliable and minimizes the risk of human errors in identifying heterozygous sites.

Related revisions were made for Results, Discussion, and Methods sections (Lines 136-137, 427-429, and 626-637).

12. Line 533-535 How were the clonal complexes (CC) numbered from goeBURST in PHILOVIZ - did they start at 0 or at 1?

Response: As per the default settings of goeBURST in PHILOVIZ 2.0, the software designated the largest CC as CC0. Subsequently, the CCs are named in descending order based on the number of isolates within each CC. We have added this statement at Lines 645-646.

13. Line 533 - “MLST clades were assigned with reference to previous publications.13, 35” How were clades defined? What program was used? If it was an in-house script, what were the criteria? The previous publications do not describe how a Clade was assigned. I do note that they both included the curator of the *C. tropicalis* MLST database. The description given in previous publications is “clades containing more than 10 genetically closely related DSTs were labelled.” The paper referred to by (13) is one that uses eBurst analysis for clonal clusters so it is unclear how clades were derived.

13 -Tseng KY, Liao YC, Chen FC, Chen FJ, Lo HJ. A predominant genotype of azole-resistant *Candida tropicalis* clinical strains. Lancet Microbe 3, e646 (2022).

35 - Zhou ZL, et al. Genetic relatedness among azole-resistant *Candida tropicalis* clinical strains in Taiwan from 2014 to 2018. Int J Antimicrob Agents 59, 106592 (2022).

Response: Thank you for the reviewer's valuable feedback. As new samples were included in this research, we conducted a thorough re-analysis of the MLST data and updated the definition of our clades accordingly.

To achieve this, we firstly constructed a maximum likelihood (ML) tree using IQ-TREE (version 1.6.12) with 1,000 ultrafast bootstrap replicates. The ML tree was based on concatenated six-locus DNA sequences from the 1,571 strains. Next, we applied a bootstrap criterion of ≥ 70 to support the determination of MLST clades, as branches with a bootstrap value of 70% or higher were considered to be well-supported and indicative of robust clades in the MLST phylogenetic tree. For phylogenetic clades that had been previously described by Zhou ZL et al. (Int J Antimicrob Agents. 2022 Jun;59(6):106592), we adopted the same nomenclature as provided in their work. For newly identified MLST clades, we designated them as Clade N1 to Clade N8.

We have updated the descriptions for MLST clades definition in the Methods section of our manuscript (Lines 639-644). Additionally, we have revised Fig. 2 that visually represents the MLST phylogenetic tree, and bootstrap values were labeled on nodes of MLST clades, providing a more comprehensive view of the tree's robustness.

14. Which MLST alleles were found to be multicopy? Long read sequencing would have made this easy to identify. How were these accounted for/handled? Was one version masked?

Response: Thanks for the reviewer's question. Indeed, we observed from the long-read sequencing results of the *C. tropicalis* reference strain MYA-3404 genome, that the *XYRI* locus used in *C. tropicalis* MLST had two copies in each haploid chromosome set. Both copies were located on chromosome 5,

with a distance of approximately 32 kb between them. Our long-read sequencing of six clinical strains also confirmed the presence of two copies of the *XYRI* gene in each strain.

Furthermore, our analysis indicated that the primers proposed by PubMLST for Sanger sequencing could amplify both copies of the *XYRI* locus in the genome. To address this challenge, we performed a separate read mapping process specifically for the *XYRI* locus by aligning all sequencing reads from each sample against one copy of the *XYRI* reference region sequences to identify variants. By adopting this approach, any SNPs occurring on either copy of the *XYRI* locus would be merged. This methodology ensures comparability of results between WGS-based and Sanger-based MLST analysis.

Additional descriptions were incorporated in the Methods section (Lines 630-636). Besides, to facilitate transparency and reproducibility, we have released our custom script described publicly available (declared at Lines 750-751).

15. Line 538-539 How was the bias of library prep accounted for? Nextera DNA prep was used. The amplification tends to bias read depth at the ends which can artificially create the impression of CNV.

Response: Thanks for the reviewer's question. We are now aware of the potential chromosome-end bias in read depth, which could introduce inaccuracies in the interpretation of CNVs. To ensure a more reliable and accurate CNV prediction, instead of using our custom script, we now employ the "Splint" algorithm developed by Gallone B et al. (Cell. 2016 Sep 8;166(6):1397-1410.e16, new Reference no. 65) for CNV analysis. This algorithm is specifically designed to address the chromosome-end bias mentioned above, and has been successfully applied in numerous previous studies involving yeast genomes. We have updated our Methods section accordingly to reflect our new workflows (Lines 667-670).

16. Line 543-544 How does this method compare to the method Ymap? Biases have not been discussed in this section. I recommend anchoring some findings of CNV by running them through the Ymap pipeline. Note the this relevant section in their publication

<https://genomemedicine.biomedcentral.com/articles/10.1186/s13073-014-0100-8>

“Several biases can impact read depth and thereby interfere with CNV analysis. Two separate biases, a chromosome-end bias and a GC-content bias, appear sporadically in all types of data examined (including microarray and whole genome sequencing (WGseq) data). The mechanism that results in the chromosome end artifact is unclear, but the smooth change in the apparent copy number increase towards the chromosome ends (Figure 2A) suggests that some DNA preparations may release more genomic DNA as a function of telomere proximity (Jane Usher, personal communication). A GC-content bias is due to strong positional variations in GC content in the *C. albicans* genome. This, combined with the PCR amplification bias introduced during sequence library or array preparation, results in a strong positional effect in local copy number estimates”
Was the method used in this manuscript also able to demonstrate the increased copy number demonstrated in the isolates by McTaggart using Ymap?

Response: Thanks for the reviewer's valuable suggestion. It aligns with the concern raised in the previous comment (Comment 15). Again, to address the potential influence of chromosome-end bias in read depth and ensure the accuracy of our CNV analysis, we have adopted the "Splint" algorithm developed by Gallone B et al, which was specifically designed to resolve the issue of chromosome-end bias.

In response to the reviewer's suggestion, we additionally utilized the Ymap pipeline to predict CNVs in five selected cluster AZR isolates, and compared the results obtained from the Ymap pipeline with those generated using our updated workflow. We found that the differences in read depth analysis results between the two methods, such as for *ERG11* CNVs, were less than 10%.

In addition, our current methodology is capable of detecting the large structural variation as demonstrated by McTaggart. However, as we clarified in our response to the reviewer's Comment 34, it is worth noting that in the report by McTaggart et al., the increased *ERG11* gene copy number observed in their strain was not attributed to tandem gene duplication mechanisms (that seen in cluster AZR *C. tropicalis* isolates). Instead, their finding resembled a mechanism previously reported in *C. albicans*, involving an additional isochromosome 5q containing *ERG11* and *TAC1* genes (Lines 189-191 and 461-471)

17. RESULTS How many new DSTs were described in the new study isolates? A description of MLST findings with a breakdown by allele is missing.

Response: In this study, a total of 726 diploid sequence types (DSTs) were identified amongst 1,571 isolates, including 201 novel DSTs that not recorded in the PubMLST database. We have added related descriptions at Lines 136-137.

18. Line 138-140 This needs to be rephrased. The data are heavily geographically biased and not representative 'worldwide'

Response: We have revised these two sentences, and removed the use of "world" to avoid making overreaching interpretations of our results (Lines 139-142).

19. Lines 141-142 Include denominator with %, so that it is clearer that the percentage refers to all the data included, >80% of which is from China

Response: As per the reviewer's suggestion, we have included denominators for all percentages presented in the relevant contents. We hope this revision ensures a more objective and transparent presentation of our results (Lines 139-144)

20. Line 144 - This is a very high rate of resistance which should be put into context of isolate selection "The overall rates of fluconazole resistance were 32.1% (n = 472) in the 1,470 isolates"

Response: Thanks for the reviewer's comment. As we addressed in our responses to the reviewer's Comment 2 above, we have now provided a detailed description of the criteria used for selecting isolates in our Methods section (Lines 568-572).

The higher azole-resistant rate released in our isolates' dataset is primarily attributed to the fact that the majority of MLST-based *C. tropicalis* studies published to date were conducted in Asia. In this region, there is a higher prevalence of azole-resistance. In addition, many studies, including our own, have focused on the molecular epidemiology of resistant populations. Taken together, these factors have resulted in an overrepresentation of resistant strains in our analysis. We have taken note of this issue and included relevant discussion in our manuscript to avoid potential over-interpretation of our results (Lines 526-531).

21. Line 149 - suggest removing the word 'led' and the word 'worldwide'. There is selection bias.

Response: As per the reviewer's suggestion, we have rephrased the entire sentence to ensure that our conclusion is not overreaching (Lines 150-153).

22. Lines 154-159 include location

Response: We have addressed the reviewer's suggestion and revised the paragraph to include geographic locations (Lines 156-162)

23. Line 180 - "aneuploidies were found on chromosome 6 (n = 5), chromosome 5 (n = 3)" - this is unexpected as aneuploidies are more commonly found on chromosome 5. What was done to check this?

Response: Each of the aneuploidy events found in this study was manually confirmed by visualizing the plot of mean read depth across non-overlapping 1-kb windows spanning the entire genome. The methodology was described at Lines 672-674. We have provided showcases of these visualizations in the revised Fig. 4.

24. Line 183 - "genome-wide high levels of heterozygosity were observed in 3.2% (17/528)" were these assessed against fungal database eg ITS database for hybrid species?

Response: As per the reviewer's suggestion, we conducted ITS region analysis for all strains exhibiting genome-wide high levels of heterozygosity. The results of the analysis indicated that there was no evidence of hybrid species among these strains.

25. Line 184-185 These needs to be rephrased and qualified "However, no associations were observed between polyploidy, aneuploidy, large structural variation or high heterozygosity genotypes and antifungal resistance."

Response: As pointed out by the reviewer in the following Comment 26, we have recognized that this statement is incorrect, particularly for isolates with Chromosome 5 aneuploidy, as it is indeed associated with the azole-resistant phenotype. As a result, we have revised the related descriptions accordingly. (Lines 189-193).

26. One would only expect these to be associated with antifungal resistance if involving Chromosome 5 as alluded to in the introduction. The only isolate in Table S2 with aneuploidy involving chromosome 5 and susceptibility data is fluconazole resistant. The other isolates with aneuploidy in chromosome 5 have susceptibility data missing. It is also not clear from Figure 4 or Table S2 which isolates were defined as having a different ploidy, and what the fluconazole susceptibility was. This data needs to be included in S2, also consider including in this paragraph.

Response: We thank the reviewer for raising this concern, and we acknowledge that our previous statements were inappropriate. In addition, as new *C. tropicalis* genomes from other publications were incorporated, and susceptibility results in Supplementary Table 2 were updated, we have re-analyzed all related data.

As a result, we have identified five isolates with chromosome 5 aneuploidy. Notably, three of these strains carry a complete extra chromosome 5, and another strain carries an extra isochromosome 5q (short arm of chromosome 5), all of which were found to be resistant to fluconazole. This resistance is likely attributed to the gain of additional copies of the *ERG11* and *TAC1* genes located in the isochromosome 5q region, a mechanism that has been well-characterized in *C. albicans* (Science 313, 367-370). In comparison, the remaining isolate that only had an additional isochromosome 5p (long arm of chromosome 5) remained susceptible to azoles.

All relevant descriptions, as well as the Fig. 4, have been revised to accurately reflect our new results, (Lines 184-186 and 189-193).

27. Line 191 - correct 'shown' as ungrammatical in this sentence "WGS phylogenetic clusters shown good correlations with MLST clades"

Response: We have modified the sentence as "Generally, association between WGS phylogenetic clusters and MLST clades was revealed, ..." (Lines 199-200)

28. Line 193 - remove 'of the' in this phrase "except for those isolates not assigned to any of the MLST clade"

Response: The sentence has been modified as suggested by the reviewer (Lines 201-202).

29. Line 194 - this requires explanation "CTC10, which was made up of isolates from MLST clades 3 and 6 that were phylogenetically divergent".

Response: The sentence has been revised, and additional details have been incorporated (Lines 202-204).

30. Line 212 - either add 'the' or change 'none of' to 'No' in this phrase "None of fluconazole-susceptible isolates"

Response: We have revised the sentence as follows: "No fluconazole-susceptible isolates involved in this study carried any of these three substitutions" (Lines 223-224).

31. Line 217-218 What does this mean? “There was also a significantly higher proportion of AZR-ADJ isolates that were *ERG11* gene-homozygous (50.0%)” Homozygosity/heterozygosity always varies across the *ERG11* gene. Is this implying that the *ERG11* gene was heterozygously present/absent? This would be unusual.

Response: Thanks for the reviewer’s comment, and we have acknowledged the potential ambiguity in our previous statement. What we would like to clarify here is that the proportion of *ERG11* A395T homozygous mutant strains is higher in group AZR-ADJ (50.0%) than in cluster AZR (7.6%). In other words, the vast majority of cluster AZR isolates harbored a heterozygous mutation A395W in the *ERG11* gene. We have revised the description to provide clearer clarification (Lines 232-2333).

32. Line 243 - This is not surprising as it is consistent with what the literature has suggested for *C. albicans* and *C. tropicalis* “Surprisingly, functional annotation indicated that this ORF was the *ERG11* gene.”

Response: We removed the word “Surprisingly” as suggested, to maintain objectivity in our description (Line 261).

33. Line 250 - “breakpoint sequences were the same in all cluster AZR isolates.” Was there any evidence of an epidemiological link between isolates? Was this a set of outbreak strains that would be expected to be clonally related? What was the SNP difference between these isolates?

Response: In this study, we found the pairwise SNP differences varied from 2,112 to 25,160 bp (median 7,731) amongst cluster AZR isolates. Our recent review (Front Cell Infect Microbiol. 2023 Mar 7;13:1130645) suggests that the SNP differences among yeast isolates from the same outbreak event are typically less than 1000 bp, with only one *Dirkmeia churashimaensis* outbreak study reporting a maximum intra-strain difference of 1621 bp. Therefore, the pairwise SNP differences observed in cluster AZR isolates far exceed this range. However, as there’s currently a lack of WGS-based investigations for nosocomial transmissions of *C. tropicalis*, it’s challenging to infer outbreaks solely from pairwise SNP differences.

It is worth noting that even when using a "lenient" threshold of 3000 bp for cluster AZR isolates, we only identified two groups of isolates (each group consisting of 2 isolates) that were obtained from the same clinical department within a short timeframe (<1 month). These findings only represented 0.6% of the 629 strains included in our WGS analysis.

Based on the results, we could conclude that the high prevalence of the azole-resistant cluster AZR strains in China does not appear to be influenced by bias due to nosocomial outbreaks. We have provided relevant descriptions in the Results section (Lines 207-208).

34. Line 264-265 This is very interesting. Are these mutations necessary for duplication events? If so what explains the increased copy number in the *ERG11* gene in the McTaggart isolates? “tandem

gene duplication events had only occurred on the chromosome with mutated *ERG11* alleles with the A395T and C461T mutations,”

Response: Thank you for the reviewer's comment. In the report by McTaggart et al., the increased *ERG11* gene copy number observed in their strain was not attributed to tandem gene duplication mechanisms, as seen in cluster AZR *C. tropicalis* isolates. Instead, it resembled a mechanism previously reported in *C. albicans*, involving an additional isochromosome 5q containing *ERG11* and *TAC1* genes. It did not carry any missense mutations in the *ERG11* gene, and only exhibited a moderate level of azole resistance (fluconazole MIC of 32 mg/L). We have incorporated relevant descriptions regarding this finding at Lines 189-191 and 461-471.

However, at present, we cannot demonstrate whether *ERG11* tandem gene duplication events and A395T/C461T mutations are essentially occurred simultaneously. Since long-read sequencing was only conducted on a limited number of isolates, we believe it would be interesting to validate this discovery in a broader range of isolates and conduct further investigations to gain deeper insights.

35. Line 285 This must be described in the methods “of the *ERG11* gene, qPCR was carried out within a subset of cluster AZR isolates (n=29)”.

Response: We have now included the description of the qPCR methodology in our manuscript (Lines 714-722). Furthermore, to offer a more comprehensive illustration of the relationship between the increase in *ERG11* gene copy number and the escalation of *ERG11* gene expression level, we have expanded the isolates for qPCR testing, and all cluster AZR/group AZR-ADJ isolates have now been tested.

REVIEWER COMMENTS

Reviewer #1 (Remarks to the Author):

I thank the authors for taking on board the feedback from the previous rounds of reviews. For my part, even though I suggested what resulted in substantial experimental work, I am happy to see it has been carried out and I'm more convinced by the results and happy to endorse

Reviewer #2 (Remarks to the Author):

It is nice to demonstrate that the drug susceptibility of isolate is dose dependent on the copy number of ERG 11. However, this data is not sufficient to support the statement "Tandem gene duplications led to high-level azole resistance in a rapidly expanding *Candida tropicalis* population". Since the authors failed to collect a drug susceptible iso-genetic progeny of drug resistant one, they may need to take a golden traditional genetic approach to generate a drug susceptible iso-genetic progeny. That is, to replace the multiple copy of ERG 11 by a single copy of wild-type ERG11.

Reviewer #3 (Remarks to the Author):

Nature Comm

This is an important contribution to the literature. The authors have extensively revised this manuscript and it is much improved. Further minor revision is required in order to clarify the following:

- * The abstract should include how many isolates were from previously reported studies downloaded from NCBI vs how many originally contributed by this study.
- * It is still not clear which isolates had heterozygous ERG11 mutations and which had homozygous. Line 232-233 now states a greater proportion of homozygous mutations in one cluster than another but this data is not presented. Table S2 should include this, rather than simply listing the ERG11 mutation. It is important as an isolate with a heterozygous Y132F mutation might be expected to have a lower MIC than one with the mutation present in the homozygous form. Was this observed?
- * ct61 is coded as susceptible FLU MIC <8 in the O'Brien paper, from their heat map and spreadsheet. The current authors have listed it as FLU MIC >8 in the table. This is important to clarify as this isolate had a Y132F mutation. If the isolate can be obtained formal anti fungal susceptibility testing (AFST) should be performed. If fluconazole susceptible it would be the first isolate that is susceptible despite this mutation, and would warrant double checking with re-sequencing. This isolate only underwent a form of agar dilution in the O'Brien paper that does not distinguish SDD isolates. Formal AFST would be important.
- * The isolates in the table below had ERG11 substitutions that should confer resistance but had fluconazole MIC 4mg/L (SDD). Some also had ERG11 CNV which the authors point out should result in a high fluconazole MIC. This does not make sense. The AFST should be double checked: Either these results should be corrected, or a possible explanation provided. Eg is there another mutation present in these isolates that is cancelling out what should otherwise cause high-level fluconazole resistance?

Position in.WGS tree	Strain ID no.	Other strain no.	WGS cluster
349	F3924	16PU1084	AZR-ADJ
374	F3972	15NX150	AZR
410	F3820	15Z2137	AZR
548	F3454	14Z2105	CTC09

* The detail provided on MLST is better. There is an explanation provided in the response for discrepancies in MLST for 10 isolates that underwent both WGS and MLST. There are also discrepancies in Table S2 between current MLST and previously reported MLST eg McTaggart. This may be due to the significant differences in methodology. Could the authors clarify

* Association with agriculture: the addition of lines 413-416 does not discuss the literature on fluconazole resistance in *C. tropicalis*. Specific MLST types have been associated with resistant *C. tropicalis* in soil, and on fruit. Is this linked to the current findings?

* Methods line 622 notes that an average read depth of 60x was considered to provide a high level of confidence for subsequent analyses. What was done with isolates with a read depth of less than this? Some downloaded data including from O'Brien had a read depth less than this..

Reviewer Comments & Author Responses

REVIEWER COMMENTS

Reviewer #1:

1. I thank the authors for taking on board the feedback from the previous rounds of reviews. For my part, even though I suggested what resulted in substantial experimental work, I am happy to see it has been carried out and I'm more convinced by the results and happy to endorse

Response: We sincerely appreciate the reviewer for dedicating the time to thoroughly review our manuscript and for endorsing our revised version.

Reviewer #2:

1. It is nice to demonstrate that the drug susceptibility of isolate is dose dependent on the copy number of *ERG11*. However, this data is not sufficient to support the statement “Tandem gene duplications led to high-level azole resistance in a rapidly expanding *Candida tropicalis* population”.

Since the authors failed to collect a drug susceptible iso-genetic progeny of drug resistant one, they may need to take a golden traditional genetic approach to generate a drug susceptible iso-genetic progeny. That is, to replace the multiple copy of ERG 11 by a single copy of wild-type *ERG11*.

Response: We are truly grateful for the reviewer's additional comments. Firstly, we are happy to learn that Reviewer 2 found the enhancements we made in our first-round revision regarding the demonstration of fluconazole susceptibility being dose-dependent on the copy number of *ERG11*.

However, the reviewer's concern persisted regarding the adequacy of our data to support 'tandem gene duplications led to high-level azole resistance in a rapidly expanding *C. tropicalis* population', and recommend to carry out gold-standard genetic approach involving the replacement of multiple copies of mutant *ERG11* gene with a single copy of wild-type *ERG11* to further validate our conclusion. Indeed, our team agrees that this experiment represents a final piece of the puzzle required to unequivocally confirm the causal relationship between *ERG11* CNV and high-level azole resistance. Unfortunately, despite our efforts over months till now, we were still unable to successfully obtain a *C. tropicalis* progeny with a single-copy *ERG11*^{WT} allele from the ancestral strains containing *ERG11*^{A395T} CNVs.

Upon reviewing published literatures, we observed that traditional genetic approaches in *C. tropicalis* were predominantly applied in industry field, primarily for metabolism research or enhancing chemical production. In addition, challenges associated with applying genetic engineering approaches in *C. tropicalis* have been acknowledged in previous studies (e.g. in the publication by Li et al., it was described that “the diploid yeast *Candida tropicalis* presents some characteristics, such as rare codon usage, difficulty in sequential gene disruption, and inefficiency in foreign gene expression,

that hamper strain improvement through genetic engineering” [1]). In the field of medical mycology research, although there have been efforts to functionally validate *ERG11* gene mutations identified in *C. tropicalis*, all such endeavors have thus far utilized the heterologous expression model within *S. cerevisiae* [2, 3] (which included an earlier work from our team). In our case, we believe the feasibility of genetic manipulation is impacted because of the unique characteristics of *ERG11* CNVs in this diploid species. The *ERG11*^{A395T} CNVs manifested in tandem repeats and situated at the same locus as the *ERG11*^{WT} allele. The significant longer segments ($n \times 3,067$ bp for each additional *ERG11* copy) of *ERG11*^{A395T} tandem repeats could reduce the efficacy of engineering approach. At the current stage, we find it challenging for our team to surmount this challenge within a reasonable expected timeframe.

References referred above:

1. Zhang, L., et al., *Development of an efficient genetic manipulation strategy for sequential gene disruption and expression of different heterologous GFP genes in Candida tropicalis*. *Appl Microbiol Biotechnol*, 2016. **100**(22): p. 9567-9580.
2. Jiang, C., et al., *Mechanisms of azole resistance in 52 clinical isolates of Candida tropicalis in China*. *J Antimicrob Chemother*, 2013. **68**(4): p. 778-85.
3. Fan, X., et al., *Molecular mechanisms of azole resistance in Candida tropicalis isolates causing invasive candidiasis in China*. *Clin Microbiol Infect*, 2019. **25**(7): p. 885-891.

We acknowledge that not fulfilling the suggestion by the reviewer may result in reduced confidence in proposing a direct causal relationship between *ERG11* CNVs and high-level azole resistance. Thus, to minimize any potential for over-interpretation of our results as the reviewer concerned, we have:

(1) Revise our result descriptions and interpretations to express our viewpoint more modestly, stating that tandem gene duplications 'contribute to' (instead of using 'led to') high-level azole resistance in the rapidly expanding *C. tropicalis* clade AZR population (in the title at Line 1 and throughout the manuscript).

(2) Acknowledge “lack of reverting the tandem duplicated *ERG11* genotype back to a single-copy allele in examining changes in azole susceptibility” as a limitation of our study in our Discussion section (Lines 534-538).

While we believe this limitation did not significantly impact the overall picture and the key message we aim to convey in this paper. Furthermore, our prior efforts in successfully establishing a dose-dependent relationship between reduced fluconazole susceptibility and an increased copy number of the *ERG11* mutant gene, through additional population analysis and *in vitro* experiments, have provided good evidence to support our conclusion.

Nevertheless, we remain committed to addressing this important issue in our future work, and will actively seek collaborations to further investigate and resolve this issue.

Reviewer #3:

1. This is an important contribution to the literature. The authors have extensively revised this manuscript and it is much improved. Further minor revision is required in order to clarify the

following.

Response: We sincerely appreciate the reviewer for reviewing our manuscript and for providing further valuable insights that to improve of our work.

2. The abstract should include how many isolates were from previously reported studies downloaded from NCBI vs how many originally contributed by this study.

Response: We've incorporated relevant information into the Abstract (Lines 39-41)

3. It is still not clear which isolates had heterozygous *ERG11* mutations and which had homozygous. Line 232-233 now states a greater proportion of homozygous mutations in one cluster than another but this data is not presented. Table S2 should include this, rather than simply listing the *ERG11* mutation. It is important as an isolate with a heterozygous Y132F mutation might be expected to have a lower MIC than one with the mutation present in the homozygous form. Was this observed?

Response: We'd like to acknowledge the reviewer's valuable suggestion. We have incorporated homozygosity information into the revised Supplementary Table S2. As suggested by the Reviewer, within the AZR-ADJ group, we observed that strains with a homozygous Y132F mutation exhibited higher fluconazole MIC distribution compared to strains with a heterozygous mutation (as shown in the figure below, with $p = 0.002$ by Mann-Whitney U test). We also included the findings in our manuscript (Lines 231-233). While azole susceptibility in clade AZR isolates influenced by CNV events, and the total number of homozygous isolates in this clade is limited. In this context, we did not conduct a further comparative analysis between heterozygous and homozygous mutant isolates within clade AZR.

4. ct61 is coded as susceptible FLU MIC <8 in the O'Brien paper, from their heat map and spreadsheet. The current authors have listed it as FLU MIC >8 in the table. This is important to clarify as this isolate had a Y132F mutation. If the isolate can be obtained formal anti-fungal susceptibility testing (AFST) should be performed. If fluconazole susceptible it would be the first isolate that is susceptible despite this mutation, and would warrant double checking with re-sequencing. This isolate only underwent a form of agar dilution in the O'Brien paper that does not distinguish SDD isolates. Formal AFST would be important.

Response: We appreciate the reviewer's query. While we believe that the fluconazole MIC for strain ct61 in O'Brien's paper should be ≥ 8 mg/L, and would like to provide further clarification.

Attached below is the original figure and its corresponding figure legend, sourced from Supplementary File (S7 Fig.) of O'Brien's paper. It revealed that the growth of strain ct61 remained unaffected in media

Editorial Note: Figure below reproduced from O'Brien, C. E. et al. Population genomics of the pathogenic yeast *Candida tropicalis* identifies hybrid isolates in environmental samples. *PLoS Pathog* **17**, e1009138 (2021).

with fluconazole concentrations ranging from 0.1 mg/L to 8 mg/L, as compared to the control group with no antifungal agents. However, inhibition of growth was observed at a fluconazole concentration of 64 mg/L. Consequently, the interpretation of this strain's MIC was ≥ 8 mg/L, which aligned with the prediction made based on the strain's *ERG11* genotype (with Y132F substitution).

S7 Fig. Phenotypic analysis of *C. tropicalis* AA isolates. 68 *C. tropicalis* isolates were grown on YPD (A) or YNB with ammonium (NH_4) (B) solid agar media as a control, and compared to strains growing on solid agar media containing different stressors. Pictures were taken after 48 hours and colony size and growth scores were measured using SGAtools [80]. Heatmaps show the normalized raw colony size in various tested growth conditions. Isolates are represented in rows, and are ordered alphabetically by strain alias. Growth conditions are shown in columns. Increased growth relative to YPD or YNB + NH_4 is shown in green (1–2) and decreased growth is shown in purple (0–1). Major differences are observed between isolates growing in the presence of cell wall stressors (calcofluor white, congo red, sodium dodecyl sulphate, caffeine), and antifungal drugs (ketoconazole, caspofungin, fluconazole). Hybrid isolates and engineered lab isolates were excluded from this analysis.

5. The isolates in the table below had *ERG11* substitutions that should confer resistance but had fluconazole MIC 4mg/L (SDD). Some also had *ERG11* CNV which the authors point out should result in a high fluconazole MIC. This does not make sense. The AFST should be double checked: Either these results should be corrected, or a possible explanation provided. Eg is there another mutation present in these isolates that is cancelling out what should otherwise cause high-level fluconazole resistance?

Position in. WGS tree Strain ID no. Other strain no. WGS cluster

349 F3924 16PU1084 AZR-ADJ

374 F3972 15NX150 AZR

410 F3820 15Z2137 AZR

548 F3454 14Z2105 CTC09

Response: We appreciate the reviewer's keen observation of these unusual results. In fact, our team had also noticed that these four strains, carrying key mutations in the *ERG11* gene, exhibited relatively lower fluconazole MICs. Notably, two of these strains belonged to clade AZR, and both carried an increased number of the *ERG11* A395T mutant gene.

To mitigate the possibility of technical errors or manual oversights, we repeated antifungal susceptibility testing for these four isolates in triplicate, and performed Sanger sequencing to validate

the mutations in their *ERG11* genes. Following these efforts, all phenotypic and genotypic results remained consistent with our initial findings.

Hence, we agree with the reviewer's suspicion, that there may be additional mechanisms contributing to the reduced resistance phenotypes exhibited by these strains. Using our WGS data, we have performed comparative genomic analysis and pan-genome analysis. Regrettably, up to this point, we have not identified specific genetic variations that could well-explain these results.

We have incorporated additional descriptions in the Results section and addressed this limitation in our Discussion section (Lines 208-209, 539-541). While regarding the primary scope of our study, these strains represent only a minor proportion of the azole-non-susceptible isolates we gathered (2.6%, 4/150) and have not significantly impacted the principal findings we aim to present in our report. Furthermore, none of these strains fall within the susceptible category for fluconazole and voriconazole; instead, all four strains were susceptibility-dose dependent to fluconazole and of intermediate susceptibility to voriconazole. These results underscore the fact that the strains still display reduced azole susceptibilities.

Given the potential scientific significance of this matter, we will continue our investigation in our upcoming research works.

6. The detail provided on MLST is better. There is an explanation provided in the response for discrepancies in MLST for 10 isolates that underwent both WGS and MLST. There are also discrepancies in Table S2 between current MLST and previously reported MLST eg McTaggart. This may be due to the significant differences in methodology. Could the authors clarify

Response: We are grateful for the reviewer's suggestion, and have included additional details in our Method section to provide better clarification of this issue (Lines 642-646).

7. Association with agriculture: the addition of lines 413-416 does not discuss the literature on fluconazole resistance in *C. tropicalis*. Specific MLST types have been associated with resistant *C. tropicalis* in soil, and on fruit. Is this linked to the current findings?

Response: We appreciate the valuable information provided by the reviewer. As suggested, we conducted a review for literatures reporting fluconazole-resistant *C. tropicalis* isolates obtained from environmental sources, such as soil and fruit. Indeed, we acknowledge that certain environmental resistant isolates belonged to MLST clade 4 and clade 5, which align with the predominant clades observed in fluconazole-resistant *C. tropicalis* isolates from humans. We have incorporated this information into our revised manuscript (Lines 424-427).

8. Methods line 622 notes that an average read depth of 60x was considered to provide a high level of confidence for subsequent analyses. What was done with isolates with a read depth of less than this? Some downloaded data including from O'Brien had a read depth less than this.

Response: We appreciate the reviewer's concern. Clinical isolates obtained from this study with a read depth of less than 60× were directly excluded from further analysis.

However, for genomes sourced from public databases and referenced in prior studies, we identified eight isolates that did not meet this read depth criterion. These included six isolates from the study by

O'Brien et al., one from the study by McTaggart et al., and one from Szarvas et al. While considering their geographic representativeness, we chose to include these strains in our subsequent work. Besides, these strains constituted a relatively small proportion of the total WGS strains investigated (6 out of 629, or 0.95%), and their inclusion was not anticipated to impact the primary conclusions drawn from our population genomic analysis.

While to enhance clarity, we have included a note in our revised Supplementary Table S2 to address this concern.

REVIEWERS' COMMENTS

Reviewer #3 (Remarks to the Author):

As noted previously, this is an important contribution and a substantial body of work and the authors are to be congratulated on the improvements to the manuscript. There is no agreed-upon methodology for fungal bioinformatics and with adequate elaboration, this may provide a template going forward. The authors have answered some of the queries but not all. Further minor clarification is required to assist readers and future researchers to understand the data presented and inform further work.

* Lines 643-646 need to be re-written. The authors have included in lines 643-646 "Of note, the manual screening for heterozygosity in Sanger-based MLST analysis could sometimes result in oversights or misinterpretation of heterozygous bases. This may potentially lead to discrepancies between the MLST results obtained through the WGS-based workflow in our study and those derived from the conventional Sanger sequencing-based approach used in previous studies for the same strains. [7,23]"

Please rephrase as the previous studies did not use Sanger sequencing, they also used WGS-based approach. The bioinformatic pipeline differed, and possible explanations may include:

- * the mapping approach particularly for XYR1 given it has been found to be multi-copy
- * Variant caller used and settings
- * Threshold for heterozygosity vs homozygosity

* The authors have included a sentence with an observation that isolates with homozygous mutations in the ERG11 gene had higher azole MICs, which is consistent with the copy number correlation. The differentiation of heterozygosity from homozygosity can be challenging as noted in the previous point. Table S2 now clearly indicates zygosity of ERG11 mutations which is very helpful. There are discrepancies with previously reported heterozygosity / homozygosity. In view of this:

- * The methods should include clarification on the threshold for calling heterozygosity/ homozygosity (which relates to the point above and may be addressed by the rephrased sentence)
- * The discussion should include as a limitation the degree of confidence that can be expressed in heterozygosity/ homozygosity, noting that it may differ with methodology, and with underlying biology - more straightforward when there is an even copy number, but as noted in S2 when there are odd copy numbers e.g. 9 or 11, dichotomously describing zygosity is more challenging

Reviewer Comments & Author Responses

REVIEWER COMMENTS

Reviewer #3

1. As noted previously, this is an important contribution and a substantial body of work and the authors are to be congratulated on the improvements to the manuscript. There is no agreed-upon methodology for fungal bioinformatics and with adequate elaboration, this may provide a template going forward. The authors have answered some of the queries but not all. Further minor clarification is required to assist readers and future researchers to understand the data presented and inform further work.

Response: We sincerely appreciate the reviewer for dedicating the time to review our manuscript again, and further revisions have been made according to the reviewer's suggestions.

2. Lines 643-646 need to be re-written. The authors have included in lines 643-646 "Of note, the manual screening for heterozygosity in Sanger-based MLST analysis could sometimes result in oversights or misinterpretation of heterozygous bases. This may potentially lead to discrepancies between the MLST results obtained through the WGS-based workflow in our study and those derived from the conventional Sanger sequencing-based approach used in previous studies for the same strains. [7,23]"

Please rephrase as the previous studies did not use Sanger sequencing, they also used WGS-based approach. The bioinformatic pipeline differed, and possible explanations may include:

* The mapping approach particularly for *XYRI* given it has been found to be multi-copy

* Variant caller used and settings

* Threshold for heterozygosity vs homozygosity

Response: Thanks for the reviewer's valuable comments. We have acknowledged that our previous explanation was inaccurate, and revised related contents in accordance with the reviewer's suggestion (Lines 648-652).

3. The authors have included a sentence with an observation that isolates with homozygous mutations in the *ERG11* gene had higher azole MICs, which is consistent with the copy number correlation. The differentiation of heterozygosity from homozygosity can be challenging as noted in the previous point. Table S2 now clearly indicates zygosity of *ERG11* mutations which is very helpful. There are discrepancies with previously reported heterozygosity/homozygosity. In view of this: The methods should include clarification on the threshold for calling heterozygosity/homozygosity (which relates to the point above and may be addressed by the rephrased sentence)

Response: We appreciate the reviewer's suggestion and have incorporated a clarification regarding the algorithm used to determine heterozygosity/homozygosity in our pipeline (Lines 621-624).

4. The discussion should include as a limitation the degree of confidence that can be expressed in heterozygosity/homozygosity, noting that it may differ with methodology, and with underlying biology - more straightforward when there is an even copy number, but as noted in S2 when there are odd copy numbers e.g. 9 or 11, dichotomously describing zygoty is more challenging

Response: We acknowledge the reviewer's concern raised. As such, we have included additional discussion addressing these limitations in our manuscript as suggested (Lines 541-544).